# What factors influence community wound care in the UK? A focus group study using the Theoretical Domains Framework

Trish A Gray,[1,2] Paul Wilson,[2,3] Jo C Dumville,[1,2] Nicky A Cullum[1,2,4]

¹Division of Nursing, Midwifery and Social Work, School of Health Sciences, Faculty of Biology, Medicine and Health, University of Manchester, Manchester, UK
²NIHR CLAHRC Greater Manchester, Salford Royal NHS Foundation Trust, Manchester, UK
³Alliance Manchester Business School, University of Manchester, Manchester, UK
⁴Research and Innovation Division, Manchester University NHS Foundation Trust, Manchester, UK

**Correspondence to**
Dr Trish A Gray;
trish.gray@manchester.ac.uk

## ABSTRACT

**Objectives** Research has found unwarranted variation across community wound care services in the North of England, with underuse of evidence-based practice and overuse of interventions where there is little or no known patient benefit. This study explored the factors that influence care in community settings for people with complex wounds, to develop a deeper understanding of the current context of wound care and variation in practice.

**Design** Qualitative focus group study using the Theoretical Domains Framework (TDF) to structure the questions, prompts and analyses.

**Setting** Community healthcare settings in the North of England, UK.

**Participants** Forty-six clinical professionals who cared for patients with complex wounds and eight non-clinical professionals who were responsible for procuring wound care products participated across six focus group interviews.

**Results** We found the TDF domains: environmental context and resources, knowledge, skills, social influences and behaviour regulation to best explain the variation in wound care and the underuse of research evidence. Factors such as financial pressures were perceived as having a negative effect on the continuity of care, the availability of wound care services and workloads. We found practice to be mainly based on experiential knowledge and personal preference and highly influenced by colleagues, patients and the pharmaceutical industry, although not by research evidence.

**Conclusions** Our study provides new insight into the role that experiential learning and social influences play in determining wound care and on the limited influence of research. Workforce pressures and limited resources are perceived to impede care by reducing patient access to services and the ability to provide holistic care. Participative collaboration between university and healthcare organisations may offer a supportive route to addressing issues, implementing sustainable changes to practice and service delivery and a resolute commitment to research use among clinical professionals.

## INTRODUCTION

People with complex wounds (open wounds, such as foot, leg and pressure ulcers, burns, open trauma and surgical wounds that are

### Strengths and limitations of this study

► This focus group study is the first to explore the factors that influence wound care and the reasons for known variation in practice.
► Employing a qualitative methodology provided new insight into the role experiential learning and social influences play in determining clinical and procurement choices.
► The focus group design stimulated discussion allowing participants to examine their own and others' views and experiences.
► The Theoretical Domains Framework provided a theoretical structure for developing a deeper understanding of wound care delivery.
► The sample was taken from community healthcare organisations in the North of England, inclusion of participants from a larger geographical population may have provided different views.

difficult to heal),[1 2] are more likely to be elderly and living with multimorbidity.[3] In the UK, the management of people with complex wounds[1 2] is mainly carried out in patients' homes or community clinics by community nurses with advice and support from specialist teams (nurses and medics with expertise in tissue viability, burns, vascular medicine or dermatology). Podiatrists also play a vital role in managing complex foot wounds, often working in conjunction with community nurses.

Care of complex wounds in community settings normally includes a comprehensive assessment of the person and their wound (involving demographics, risk factors for wound healing, quality of life measures, wound status, wound parameters and symptoms), specific wound-related assessments such as Ankle Brachial Pressure Index (ABPI) for people with venous leg ulcers and implementation of appropriate interventions.[4] Interventions may involve wound cleansing followed by dressing to manage exudate and

protect the wound. While dressings are used widely across wound types, with many different options available, there is currently no evidence that one dressing type is more clinically or cost-effective than another, even in the case of relatively expensive antimicrobial dressings. In contrast, there are effective first-line treatments which should be widely used, such as the use of compression therapy for venous leg ulceration which is known to reduce time to wound healing.[5 6]

As part of a wider programme of wound care research funded by the National Institute for Health Research Collaboration for Leadership in Applied Health Research and Care Greater Manchester (NIHR CLAHRC GM) , we conducted a survey to assess how healthcare professionals managed wound care across five community healthcare organisations in the North of England.[7] The findings are discussed in more detail elsewhere[7] but in summary the survey revealed unwarranted variation in clinical practice, with general underuse of Doppler-aided measurement of ABPI,[8] underuse of compression therapy and[9] potential overuse of antimicrobial dressings.[6] In the UK, variations in wound care are being recognised and addressed with initiatives such the Leading Change, Adding Value Nursing and Midwifery Framework[10 11]; however, there has been little formal exploration of drivers for this variation in the delivery of wound care and barriers to implementing the findings from current research evidence. In turn, there is little intelligence to guide further research implementation and bring about meaningful practice change with the aim of maximising patient benefit.

Our aim was to identify and explore factors that influence care in community settings for people with complex wounds. We wanted to better understand the current context of community wound care and how research evidence informs care delivery.

## METHODS
### Design
We conducted six focus group interviews to explore the factors that influence the care of people with complex wounds in community settings. The Theoretical Domains Framework (TDF) was used to structure the questions, prompts and analyses.[12 13] The TDF provides a theoretical lens through which to view cognitive, affective, social and environmental factors that could potentially influence behaviour.[14] It has been used extensively across a range of clinical areas.[15–17] Its constructs are grouped into 14 discrete domains.[12] The TDF is presented in table 1 showing the domains, definitions and examples of behaviours related to wound care and wound product procurement.

### Participants and settings
Purposive sampling was used to ensure that we recruited participants with relevant clinical and/or procurement experience. Eligibility included community-based clinical professionals who cared for patients with complex wounds or non-clinical professionals who were involved in the procurement of wound care products. Clinical professionals included community nurses, podiatrists, tissue viability or burns specialist nurses, wound research nurses and clinical nurse managers (who had a clinical role, managed a team of community nurses and were responsible for wound product procurement decisions). Non-clinical professionals included: medicines optimisation pharmacists, procurement leads, procurement advisors and medicines management leads. There were five multidisciplinary focus group interviews for clinical professionals; one for each participating provider organisation. Four were drawn from provider organisations in one defined geographical area with a fifth conducted in a different geographical area but similar urban conurbation in the North of England. The latter was chosen for its well-established links with university researchers as a comparison to the other organisations where collaborative partnerships with university researchers were in their infancy. A separate focus group interview was held for non-clinical professionals. As the themes for clinical and non-clinical focus group interviews differed, we chose to separate clinical from non-clinical professionals to maintain focus and create an optimum environment for free flowing discussions. Potential participants were identified through contacts developed as part of the NIHR CLAHRC GM wound care programme and were approached via email, telephone or face-to-face meeting. Focus group interviews were held locally to participants' work place in a healthcare setting or conference centre.

As participants were drawn from a relatively homogeneous population and the interview schedules were focused on specific aspects of wound care and wound product procurement, we anticipated that we would reach data saturation within three to four focus group interviews; however, to incorporate all partner provider organisations using the format described above we needed to recruit 50–60 participants in total across the six groups (to allow for 8–10 participants per group), based on recommendations from existing literature.[18–21]

### Data collection
The format was similar for all focus group interviews; they were facilitated by a lead (TAG) with one or two co-facilitators (PW and JCD). All facilitators were experienced researchers and familiar with the evidence base for wound care. A fourth member of the research team took field notes. Before the session began, participants were asked to complete a brief demographic questionnaire to clarify their academic and professional qualifications and wound care/product procurement experience as relevant; these data were used to describe the participants involved and were not linked to particular responses or quotes. Each session was audio-recorded with recordings deleted following verification of anonymised transcripts.

**Table 1** The Theoretical Domains Framework: domains, definitions and examples of behaviours related to wound care and wound product procurement

| Domain | Definition | Examples of wound care and wound product procurement behaviours |
|---|---|---|
| Knowledge | An awareness of the existence of something. | Knowledge of wound types, wound aetiology, risk factors, wound product types. Wound knowledge is influenced by education, experience and research. |
| Skills | An ability or proficiency acquired through practice. | Ability to complete a comprehensive wound assessment, specific assessments such as Ankle Brachial Pressure Index, apply compression bandages/stockings, manage procurement processes effectively. |
| Social/Professional role and identity | A coherent set of behaviours and displayed personal qualities of an individual in a social or work setting. | Carrying out a clinical or procurement role according to job description, communicating and working appropriately and effectively with other clinical or non-clinical professionals. |
| Beliefs about capabilities | Acceptance of the truth, reality or validity about an ability, talent or facility that a person can put to constructive use. | Confidently making the right decisions about care for patients with complex wounds, confidence in negotiating skills for product procurement. |
| Optimism | The confidence that things will happen for the best or that desired goals will be attained. | Confidence that care provided will cure/manage wounds effectively, confident that most cost-effective products can be purchased. |
| Beliefs about consequences | Acceptance of the truth, reality or validity about outcomes of a behaviour in a given situation. | Having realistic views about patient adherence to treatment plans and healing rates for complex wounds. |
| Reinforcement | Increasing the probability of a response by arranging a dependent relationship, or contingency, between the response and a given stimulus. | Support of colleagues, team work, wound care provided has produced the desired goal, research evidence that interventions work. |
| Intentions | A conscious decision to perform a behaviour or a resolve to act in a certain way. | To practice according to a care plan, national and international guidelines. |
| Goals | Mental representations of outcomes or end states that an individual wants to achieve. | Setting goals for wound healing, improving patient adherence, achieving competence for a new skill. |
| Memory, attention and decision processes | The ability to retain information, focus selectively on aspects of the environment and choose between two or more alternatives. | Ability to remember wound care information, dressing specifications, considering the wide choice, making decisions based on evidence. |
| Environmental context and resources | Any circumstance of a person's situation or environment that discourages or encourages the development of skills and abilities, independence, social competence and adaptive behaviour. | Organisational structures, procedures and processes, workload pressures, staff shortages, funding constraints, service cuts, procurement processes, product cost, product availability. |
| Social influences | Those interpersonal processes that can cause individuals to change their thoughts, feelings or behaviours. | Decisions influences by personal, colleagues' patients, pharmaceutical industry preferences, team work and shared care, understanding patients' needs, negotiating product cost. |
| Emotion | A complex reaction pattern, involving experiential, behavioural and psychological elements, by which individual attempts to deal with a personally significant matter or event. | Coping with wounds that do not heal, managing challenging wounds, dealing with emotions related to patient morbidity and mortality. |
| Behavioural regulation | Anything aimed at managing or changing objectively observed or measured actions. | Formulary to guide/monitorprescribing and procurement choices, audits of practice and procedures. |

Adapted from Cane *et al.*[12]
Licensee: BioMed Central.

## Procedure

The discussion explored specific behaviours linked to the TDF domains and reactions to site-specific, regional and national procurement data using the questions and prompts outlined in online supplementary appendix 1. Clinical professionals were encouraged to think about factors that from their experience, enable or hinder the delivery of wound care, relating their answers to their own experiences. Through prompts we probed further, allowing participants' reactions to unfold, giving them the opportunity to explore their own and others' views. We continued to prompt if responses were not spontaneously

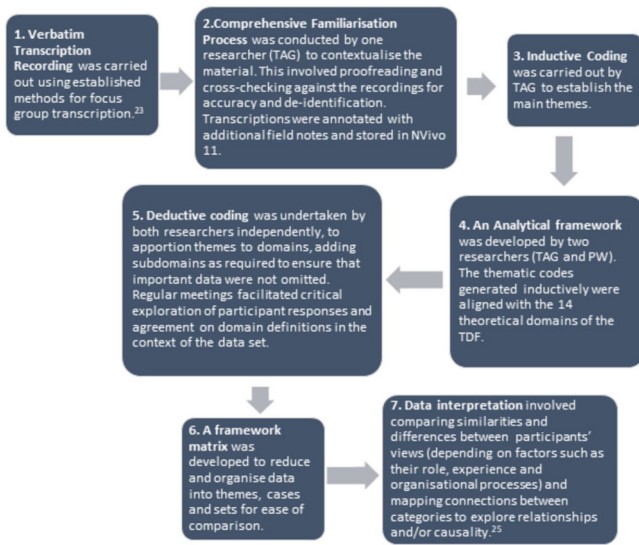

**Figure 1** Qualitative analysis using a seven-step framework method.

offered to encourage full participant engagement. The focus group interview for non-clinical professionals followed an identical format with the questions more related to procurement systems and procedures (online supplementary appendix 2). Interview schedules were piloted by specialist nurses, clinical managers and a procurement lead, after which minor amendments were made. Respondents validated the accuracy and completeness of the findings[22] following a verbal summary (taken from the field notes) at the end of each focus group interview and a post-analysis report sent via email.

**Patient and public involvement**
Views expressed by members of the NIHR CLAHRC GM Wounds Research PPI Forum about their experiences with healthcare professionals and wound care services were used to inform some of the questions and prompts for the focus group interviews.

**Data analysis**
Quantitative data were stored in SPSS (IBM v.22). Demographic variables are expressed in frequencies, means and SD where distributions are normal, and medians and range when skewed. Qualitative analysis followed a seven-step process in line with the framework method (figure 1).[23–25]

**FINDINGS**
**Participant characteristics**
Sixty participants were invited to attend one of six focus group interviews (mean duration: 106 min). Fifty-four participants attended while nine invited participants could not attend due to other clinical commitments or annual leave (three of whom nominated colleagues to attend in their place). Participants comprised 46 clinical professionals (10 specialist nurses (19%), 25 community nurses (46%), 7 podiatrists (13%), 3 clinical managers

| Table 2 | Participant characteristics (n=54) | |
|---|---|---|
| **Gender** | | |
| Male | 7 (13) | |
| Female | 47 (87) | |
| **Role group clinical professional** | | |
| Specialist nurse | 10 (19) | |
| Community nurse | 25 (46) | |
| Research nurse | 1 (2) | |
| Clinical manager | 3 (5) | |
| Podiatrist | 7 (13) | |
| **Role group non-clinical professional** | 8 (15) | |
| **Highest academic qualification** | | |
| MSc | 6 (11) | |
| BSc/BA (Hons) | 27 (50) | |
| PG Diploma | 11 (20) | |
| PG certificate | 2 (4) | |
| Vocational qualification | 5 (9) | |
| A level | 3 (6) | |
| **Years in current role** | **Mean (SD)** | |
| Clinical professional | 8.6 (7.4) | |
| Non-clinical professional | 4.7 (4.3) | |
| **Years of wounds care/procurement experience** | | |
| Clinical professional | 14.5 (8.8) | |
| Non-clinical professional | 5.7 (6.4) | |
| **Attended a wound care update in last 12 months (n=46)** | **N (%)** | |
| Yes | 15 (33) | |
| No | 31 (67) | |
| **Attended wound procurement update in last 12 months (n=8)** | | |
| Yes | 1 (13) | |
| No | 7 (88) | |

(5%) and 1 research nurse (2%)) and 8 non-clinical professionals (15%). Wound care experience was extensive (mean 14.6 years, SD 8.8) among clinical professionals (table 2).

**Key themes identified within relevant domains**
Five TDF domains dominated: *environmental context and resources*, *knowledge*, *skills*, *social influences* and *behaviour regulation*. The domains of knowledge and skills were closely linked and frequently overlapped, therefore, we combined these. We did not code any source data to the domains of *emotion* and *intentions* and found the remaining six domains to overlap with the five dominant domains. We have therefore, focused on the five key domains which best explain the variation in wound care and the underuse of research evidence. The coding

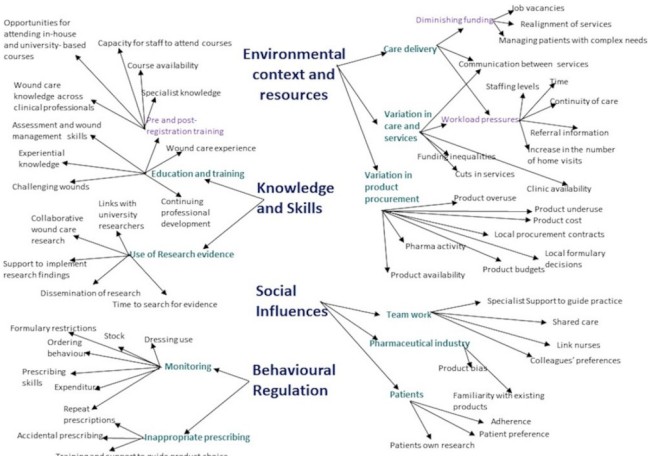

**Figure 2** Coding tree showing the four salient domains with connected subthemes.

tree (figure 2) demonstrates the relationships between domains and subthemes.

## Environmental context and resources
### Delivery of care

Clinical professionals across all groups expressed feeling the pressure of increased workloads. Some participants said they were working more intensely and without breaks, constantly feeling anxious that they may have missed something as time was limited between patient consultations. They reported that there was an increase in sick leave, experienced colleagues were leaving and their roles were left vacant.

> You haven't got the same skill base any more. We haven't got the same expertise, we're losing our experienced link nurse this week, and we haven't really got anybody with that level of skill in wounds to take her place…we've got 30 vacancies at the moment that haven't been filled. (Clinical manager)

Community nurses reported that specialist clinics were being cut, and patients previously seen in dedicated leg ulcer clinics by nurses with specialist knowledge, were now visited at home by understaffed community nursing teams.

> Physically running the clinic was based on when there was about six or seven (leg ulcer specialist) staff. … when it was a leg ulcer service. There's only two of us so we haven't got the capacity to cover those let alone do all the home visits. (Community nurse)

Community nurses and podiatrists voiced concern that undue time was spent gathering required patient information due to poor referral information supplied by hospital staff.

> You constantly are ringing because they'll (ward staff) put (on the referral) 'care of wound', but… what wound have they got? What operation have they had? What would you like me to do with it? It's very, very poor. (Community nurse)

## Variation in care and services

Many clinical participants attributed variation in the patterns of care delivery to realignment of services due to reduced funds. The majority of clinical professionals reported that specialist leg ulcer clinics had been cut resulting in a greater number of home visits for community nurses. Participants from the research active organisation reported that practice nurses (nurses based in a general practitioner practice providing primary care for a local population) managed mobile patients with wounds, while community nurses cared for housebound patients with more complex health needs. This changed model of service delivery was felt by the community nurses to have eased their workload.

Participants from organisations that managed both hospital (acute) and community services felt that resourcing prioritised the acute service at the expense of the community service. Participants made reference to the differences between resources available in acute care that were limited or unavailable in the community; this included wound care products and digital technology.

> I just don't feel the acute side has got a grip at all on community services in terms of what we do…I mean, I do a specialist (acute) clinic on a Tuesday morning and have access to all sorts of dressings. And I come back into the community….and we're very limited, we've got one foam (dressing) that we can use. (Podiatrist)

Clinical participants viewed access to photographic equipment as a valuable resource that allowed images of wounds to be sent to a podiatrist or specialist nurse for rapid diagnosis and care planning, however, only healthcare professionals with access to hospital photographic equipment could make use of this service. One specialist nurse apologised for using photographic equipment that the community nurses within her organisation did not have access to.

> I do (take photographs of wounds). I've got a camera. Sorry. It is downloaded onto a programme at the hospital. So that's probably why (I have access to it). (Specialist nurse)

## Variation in product procurement

Participants reported a variety of wound care product procurement processes; some (across two provider organisations) obtained all products via prescription, others (across two provider organisations) used a combination of prescribing and stock purchase and one group (one organisation) operated a total stock purchase system. All participants noted the local use of wound care formularies (a locally developed list of recommended products), to guide prescribing or purchasing decisions,[26] however, through discussion it was recognised that the products listed and the number of product available varied across formularies. One organisation had a very restrictive formulary and monitored use closely; participants found

this restrictive formulary enabled them to choose appropriate products.

> I think it's an enabler, … there are so many (dressings to choose from) you can go completely for something that costs so much and something that wouldn't be right … but having that formulary means that we know what we can choose. (Community nurse)

### Knowledge and skills
#### Education and training

All nurse participants agreed that there was a limited amount of wound care education for student nurses and that most wound care knowledge and skills were gained through community placements rather than in the classroom. Only specialist nurses had attended a university-based post-registration wound care course. In contrast, podiatrists received regular undergraduate and postgraduate wound care education. All clinical professionals viewed wound care knowledge across other services (hospital, primary care and nursing homes) to be poor, which increased their workload if aspects of care, documentation or prescription information were incomplete.

Specialist nurses reported that due to workforce pressures, in-house courses they offered were often cancelled or attendance was poor. Due to these difficulties, specialist nurses relied on the pharmaceutical industry to provide wound product training sessions. Concerns were raised particularly by the non-clinical professionals that the educational contributions of industry representatives were highly likely to favour their own products.

> But then at the same time they'd spy the competition and they'd basically suggest that their products are equivalent to those products that were already on the shelf …….and then we were inundated with requests for new products. (Non-clinical professional)

#### Use of research evidence

Only participants from the provider organisation with a history of collaborative wound care research indicated that they actively sought to keep up to date with research. Specialist nurses from this focus group talked about their established links with university researchers and their involvement in co-producing wounds research with academics. They discussed disseminating relevant research findings through electronic newsletters, workshops and meetings with community staff and where capacity allowed, staff were supported to implement research findings. Participants reported that their organisation was highly research active in wound care; clinical professionals had participated in research that found compression stockings to be more cost-effective than compression bandages for people with venous leg ulcers[9] and they subsequently implemented the findings into practice. The remaining participants viewed research with caution, they found very little time to search for evidence or be involved in research.[6 27 28]

> I can't know everything about all dressings, and therefore you often stick to what you know and you don't often have time to look at research. (Non-clinical professional)

> And as healthcare professionals it's not built into our contracts to do research…there's no time put aside. (Specialist nurse)

### Social influences
#### Team work

The importance of good teamwork was frequently emphasised and acknowledged by all participants. Much of the sharing of experiences was conducted informally. Clinical professionals reported that team support alleviated some of the current workload pressures and shared care was viewed as a valuable method for joint decision making. Participants from one focus group only reported the existence of wound care link nurses whose role was to cascade new information, new research evidence and product updates from specialist nurses to their colleagues. However, capacity issues were affecting the scope of this role.

#### Industry and patient influence

As referred to above, all participants were concerned about the influence that the pharmaceutical industry had on product choices. It was felt that this influence varied depending on how closely pharmaceutical representatives' access was monitored. Non-clinical participants were particularly and negatively vociferous about the influence of pharmaceutical representatives yet viewed the role of policing any promotional activity as a specialist nurse responsibility.

> You can police this a little bit more in acute, can't you, but in the community we were fighting a losing battle with the reps when they're just given free range to provide training. (Non-clinical professional)

Participants were very aware of the influence that patients have on wound care, which at times caused difficulty finding a suitable dressing that met patients' expectations. Participants reported that some patients removed dressings earlier than necessary if minor staining appeared. Participants were mindful that careful assessment and monitoring patients' adherence to therapy was necessary when making product choices. Participants also found that patients searched for information on the internet in an attempt to influence product decisions.

> She (the patient) read that honey was good and she thought I'll go and buy my own…. and swore it did the trick, so who are we to argue with her? (Community nurse)

### Behaviour regulation

Community nurses reported that antimicrobial dressings (particularly silver-impregnated dressings) were used for individual patients for a 2-week trial period and then

reviewed, however, they acknowledged that if use was not closely monitored there was potential for overuse. Non-clinical professionals and specialist nurses were aware of the high expenditure on antimicrobial dressings but acknowledged difficulties in monitoring effectively and providing adequate training and support due to capacity issues.

> Silver spend is still a problem and it's a long-term…
> …I think there's still habitual use, district nurses having the time to stop and think and review and stop a treatment rather than continue. (Specialist nurse)

Specialist nurses reported that general practitioners regularly prescribed high cost antimicrobial dressings for nursing home residents. The prescription for these dressings would often be repeated without review unless the resident was referred to a specialist nurse. There was an opinion among participants that some prescribing of silver dressings may be accidental because dressings are listed alphabetically in some prescribing platforms and silver dressings appear first (as they are denoted by the chemical symbol for silver, 'Ag').

## DISCUSSION

We believe this is the first study to explore factors influencing care in community settings for people with complex wounds while seeking to understand the reasons for known variation in practice. Overall, participants described a challenging working environment, with influences such as workforce shortages and diminishing treatment resources having a marked effect on continuity of care, patient access to services and workload. Clinical practice seemed to be predominantly based on experiential knowledge, personnel preference and to be highly influenced by colleagues, patients and the pharmaceutical industry.

### Workforce pressures and diminishing resources

Wound care services were described by participants as a working environment characterised by increasing time pressures and diminishing resources. Roles were perceived as becoming task orientated which was felt to dilute the quality of care. Participants reported there was a rise in sickness, colleagues were leaving for less pressured roles and vacancies were not being filled. UK surveys of community nursing services have found similar results.[29–31] The UK has fewer nurses relative to the population than many EU countries.[32] The number of community nurses is falling, with an estimated vacancy rate of 9.4%.[33] Forty per cent of experienced nurse positions are vacant.[34] Championing flexible career pathways, integrated care and the introduction of combined hospital and community posts (to standardise practice, improve care coordination and vary work experiences) have been proposed by UK governing bodies to improve retention rates.[35–38]

Participants reported that specialist clinic sessions had been cut, resulting in increasing workload pressures for community nurses. A systematic review of 27 studies found improved information technology, including remote specialist consultations, to improve access to specialist input, provide educational support for the referrer, shorten referral time and avoid unnecessary travel and inappropriate visits.[39]

### Experiential learning and social influences

All nursing participants agreed that there was a lack of wound care education in basic nurse education. Wound care skills were learnt during community but not hospital placements. This was verified by participants' reference to insufficient information from hospital nursing and medical staff on referral forms and via telephone calls which cause delays in assessment and frustration for community nurses. While all specialist nurse teams offered ongoing wound care training to community nurses, cancellation or poor attendance frequently occurred due to staff shortages. By contrast, podiatrists' viewed their wound care education to be strong before and after qualification.

In the light of the current workforce issues and the difficulties community nurses had updating their wound care knowledge, other strong influences played a significant role in wound care choices such as personal, colleague and patient preferences as well as the influence of pharmaceutical company representatives. This influence can drive variation in dressing and treatment choice depending on the amount of access pharmaceutical representatives have to healthcare settings and clinical professionals' attitude to the information they provide.[40] The ongoing cuts to continued professional development funding in the UK since 2015 may lead to greater dependence on the pharmaceutical industry for training and 'education', which is problematic due to companies' vested interests in the use of specific products.[41 42] Interprofessional education may break down professional boundaries and provide opportunities for mutual learning and joint solutions across professional groups and specialties.[43] Further investment into evaluated training interventions that are of high quality and independent is warranted to ensure education is consistent and effective; providing healthcare professionals with the confidence to make the right decisions to improve continuity and quality of care.[36 44–46]

### The influence of research on wound care
#### Using research to guide product choice

There is a plethora of wound products available for use but, as several Cochrane systematic reviews have shown, there is a paucity of research evidence showing that products are clinically effective.[6 47–52] Despite this, product use and expenditure have grown; particularly antimicrobial dressing use, where no compelling evidence or guideline recommendations exist to support routine use (Hussey *et al*, 2018, manuscript submitted for publication).

We found that a restrictive formulary was viewed as enabling better patient management, particularly if guidelines accompanied the formulary. Community nurses found a formulary and guidance gave them more assurance that they were making the right decisions and specialist nurses found formularies reduced inappropriate product choices and assisted in standardising product use across their service. For the majority of organisations, however, the formulary acted as guidance only and 'off-formulary' prescribing could occur without restriction unless resources were available to monitor prescribing behaviour closely. National guidelines exist to guide the use of specific products,[4 6 27 28] however, national standards to guide choice across the range of wound care products would reduce variation of product use and guide more rational prescribing.[53]

### Engagement in research

Research was raised as a factor influencing wound care in only one, highly research-active provider organisation. In this site, well-established links with university researchers had been highly influential. Current evidence suggests that there is an association between the engagement of individuals and healthcare organisations in research and improvements in healthcare performance.[54] In the other sites, where collaborative links with university researchers were more newly established, research informed decision making was more limited and research generally was viewed with caution. Much of the discussion around acquiring knowledge and skills to inform wound care decisions was related to experiential influences; day-to-day wound care experience, watching others and consulting with more experienced colleagues and specialists. This finding is in line with other research showing that experiential learning and the social influence of peers rather than research knowledge are major influencers on nursing practices.[55–57]

If evidence obtained from research is to inform management and practice, robust, long-term strategies to support and facilitate its use will be required. In England, the NIHR funded research that incentivises co-production of research, for example, NIHR CLAHRCs represent an ongoing nationwide experiment to close the distance between research production and research use.

### Limitations

The main limitation is the sample which was taken from community healthcare provider organisations in the North of England and included only one research-active organisation. Inclusion of participants from a larger geographical population may have provided different views, however, we captured many of the issues affecting healthcare (such as work pressures, staff shortages and limited resources) across the UK[30 58 59] and further afield[60 61] due to the financial healthcare crisis worldwide. We would have preferred to include more than one research-active organisation but due to the limited number of research-active organisations within our geographical area as well as funding and time limitations we could not recruit more. We were able to recruit the recommended number of participants for each focus group but work pressures dictated the range of clinical professionals and for one group there were no podiatrists which may have reduced the diversity of views, for that particular interview. However, there was good representation from podiatrists across the other groups ranging from senior management to junior positions. Only one research nurse was able to participate and as the research-active organisation was the only organisation to employ a small team of wound research nurses it is not surprising that we could only recruit one.[30 58–61]

A challenge of using the TDF was the overlap across domains such as *knowledge* and *skills, beliefs about consequences* and *social/professional role and identity*. Other authors have reported similar issues.[17 62] The recently published guide to using the TDF addresses these and other challenges to promote the use of the TDF to a wider audience.[14]

Finally, our aim in this study has been to surface factors that could potentially explain variations in the delivery of wound care. We of course recognise that wound care is complex and multifaceted involving a wide range of behaviours. Given this, we recognise that any formal attempts to develop strategies to modify existing practices and behaviours will require a level of granularity beyond what is available in the data presented. Our study does shed light on those domains where those future efforts should focus.

### CONCLUSIONS

Our study provides new insight into the role experiential learning and social influences play in determining management and treatment choices and on the limited influence of evidence obtained from research. Workforce pressures and limited resources were perceived by the participants to impede care by reducing patient access to services and the ability to provide holistic care. Co-production of research evidence through participative collaboration between university and healthcare provider organisations may offer a supportive route to addressing issues, implementing sustainable changes to practice and service delivery and a resolute commitment to research use among clinical professionals.

**Acknowledgements** The authors would like to thank many colleagues within the NIHR CLAHRC Greater Manchester Wound Care Programme, who have assisted with and supported this work. The authors would like to thank all participants for giving their time, sharing their views and for their enthusiasm throughout.

**Contributors** NAC, JCD and TAG conceived the idea and design for the overall project. PW contributed to further development of the study design. TAG, PW and JCD collected the data. TAG and PW were responsible for data analyses. TAG created the original draft of the manuscript. All authors contributed to the interpretation of study findings, critical revision of the manuscript for important intellectual content and approval of the final manuscript.

**Funding** This project was funded by the National Institute for Health Research Collaboration for Leadership in Applied Health Research and Care (NIHR CLAHRC) Greater Manchester. The NIHR CLAHRC Greater Manchester is a partnership

between providers and commissioners from the NHS, industry and the third sector, as well as clinical and research staff from the University of Manchester.

**Disclaimer** The views expressed in this article are those of the authors and not necessarily those of the NHS, NIHR or the Department of Health.

**Competing interests** None declared.

**Patient consent for publication** Not required.

**Ethics approval** Ethics approval was sought and granted from the University of Manchester Research Ethics Committee (Refs 15272, 15327 and 2017-0559-1767) and HRA approval was sought and granted (Refs IRAS 174691, 184865 and 219918). Written informed consent was obtained from all participants

**Provenance and peer review** Not commissioned; externally peer reviewed.

**Data sharing statement** Requests for access to data should be addressed to the corresponding author.

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
