## [Reviewer comments · BMJ Open]

ARTICLE DETAILS

TITLE (PROVISIONAL)	WHAT FACTORS INFLUENCE COMMUNITY WOUND CARE IN THE UK? A focus group study using the Theoretical Domains Framework
AUTHORS	Gray, Trish; Wilson, Paul; Dumville, Jo C.; Cullum, Nicky

VERSION 1 – REVIEW

REVIEWER	Souraya Sidani Ryerson University, Canada
REVIEW RETURNED	28-Jul-2018

GENERAL COMMENTS	The authors report on a qualitative study that examined factors perceived by healthcare professionals to influence their wound care practices in community organizations. The study appears to build on and extend a prior one that identified what seem to be less-than-optimal practices, and is appropriately guided by the Theoretical Domain Framework (TDF). Although the main research topic is of interest, there are several points that require clarification and more consistency in reporting across sections. The introduction is very brief and does not clearly situate the study within the authors' program of research and within related literature. - The definition of complex wounds is not easy to understand by readers unfamiliar with wound care.- It would be helpful to clarify the role of nurses and specialist team members in wound care, and to clearly identify the population targeted in this qualitative study.- It is essential to explain what aspects or practices related to wound management were investigated in the previous study done by the authors, and the current study: was the focus on general wound care practices, on evidence-based guidelines, or just the use of equipment like the doppler? Once the practices of interest are identified, then it would be useful to provide more information on variation in its implementation as found in the previous study and to relate the TDF factors to these particular practices. Further, more information is needed on what is considered best practices and how the previous study findings deviate from these best practices – to put the current qualitative study in context.- Is the aim of the study to examine wound care management and prescribing practices? Are these two different types of practices? How is each defined? The findings (related to factors influencing practices) presented do not allude to any specific practice. The Methods section lacks explicit justification for the methodological decisions made.
---

	 - What is the target population for this qualitative study: only nurses and/or other members of the team responsible for wound management or for prescribing? Prescribing what: medical treatments or wound related materials? Were participants in this qualitative study selected from those who took part in the previous quantitative study? - What were the community settings targeted for the study: primary care, community care, ambulatory care or home care? - What were the different professional groups included in the group interviews? Should the results then be presented by professional groups instead of being lumped regardless of professional groups? Or, were the emerging themes the same across professional groups? This point should be clarified in light of the limitation stated that one podiatrist participated in the study. - Why were clinical nurse managers included? Do they provide direct patient care? - Please, provide explicit recommendations for sample size in qualitative research; this is important to determine the adequacy of the accrued sample size in total and for each professional group. - Page 6, line 45-46 – the data do not “summarize” but “describe characteristics” of participants. Please be careful with terminology. - What specific barriers were discussed? - What strategies were used to maintain reliability and validity in this qualitative study? The findings section could be improved.  - It would be helpful to summarize the main characteristics of participants in the text, rather than just referring to the table. - It is a bit difficult to follow / understand the findings or themes that emerged:  1. It is unclear what specific practices / behaviors were discussed, and were found to be influenced by the factors listed. 2. It is unclear if the findings / identified factors were mentioned by all professional groups across all organizations, or if some factors were more relevant to some groups in some organizations, which is expected under different environmental contexts. 3. The quotes presented are not necessarily consistent with or do not clearly illustrate the respective theme. For some themes, no quotes are presented. 4. There is no explanation of how exactly the identified factors influence what particular practice. 5 It is not always clear what each factor is exactly about and codes reflect it. - The authors may want to consider replacing the table listing the questions asked to explore each theoretical domain by a table that summarizes the domains, factors that emerged from the qualitative data analysis and related to each domain, the specific codes reflecting the factors, and the most illustrative quote. The discussion appears to be selective, addressing specific factors (some of which were mentioned very briefly in the results section, such as the issue of technology) instead of the identified domains – this gives the impression of inconsistency across sections.
REVIEWER	Amber Young University Hospitals Bristol NHS Foundation Trust, UK I am currently working with Ms Jo Dumville on a paper on which I am the senior author and she is the co-author. She is also on the steering group of another NIHR project that I lead on. I am happy that my review is unbiased.

GENERAL COMMENTS

This study explored the factors that influence community wound care to develop a deeper understanding of this variation at the patient, the service and the organisation level.

General comments:

The subject matter is important and interesting. The paper is well written, informative and reads well. There are some issues, however, that impact on generalisability. The study was carried out in only one geographic area and with only one research-active organisation. I think the authors could have discussed this in a little more detail in the discussion / limitations and justified why this methodology will not limit generalisability within the UK in their view. This is the main limitation of the study. The introduction needs more detail in my opinion on this study rather than a previous study, including detail of wound care pathways and evidence for existence of variation in practice ie the basis for why this study is needed. The themes reported in the results are a mix of information from participants and facts. It is sometimes unclear which was which. Many of the participant views are not backed up by facts and it is unclear if these are unchecked views or views based on checked facts?

Detailed comments:

Abstract: ability of provide holistic care: typo

Organisations need to develop strategies to apportion resources wisely and redeploy skills: I do not think this can be concluded from this study and is not in the main conclusion?

Introduction:

....open trauma and surgical wounds: what is the definition of a complex wound – these seem to be wounds that should heal simply? Are burns included?

Needs more detail on wound care pathways and also evidence for variation in practice and need for this study?

Methods:

Purposive sampling was used to ensure that we recruited participants with relevant community clinical, management and procurement experience.: more detail please ie what is relevant?

The focus group interviews were arranged by professional group: No multidisciplinary groups – why was this decision made
.....regularly cared for patients with complex wounds: what does regularly mean?

The NIHR CLAHRC Greater Manchester Wounds Research PPI Forum provided views on their experiences with healthcare professionals and wound care services.: this is unclear – how were patients and public involved?

Specialist nurses, clinical managers and a procurement lead were involved in piloting the interview schedules, after which minor amendments were made. Member validation was performed at the end of each focus group interview and following analysis: how does this relate to PPI and what is member validation please?

No corrections were requested.: why was this?

	Results: The authors state purposive sampling was undertaken – but also state that 87% of participants were female participants and only one research nurse?? I think this requires more explanation of justification. As referred to above, the pharmaceutical industry is also influential; providing training and education around product use, although this varied across organisations depending on the capacity of specialist nurses to limit access and monitor sessions: where is this information from? In order to standardise and better regulate prescribing behaviour across several provider organisations including primary care, plans were in progress to produce a regional formulary: where is this information from? For this reason, only obligatory monitoring appeared to be conducted.: fact or comment? Participants across all focus group interviews expressed feeling the pressure of increased workloads and Another concern that caused unnecessary stress and treatment delays was the poor referral information received from hospital medical and nursing staff for patients discharged to the care of community nurses (p11): was there any evidence for this and other findings from different sources? Three community services had amalgamated with hospital (acute) services in the last five years and this was reported to have had a profound effect on staff morale.: what effect? In terms of training for post-qualified nurses, university-based wound care courses were available but these were mainly accessed by nurses preparing for a specialist wound care role and were not often attended by generalist nurses, resulting in a limited number of nursing staff with a higher level of wound care knowledge. Participants viewed wound care knowledge across other services to be particularly poor especially amongst hospital and nursing home-based nurses and GPs.: is the first part fact? Only those participants from the healthcare organisation with a history of collaborative wound care research indicated that they actively sought to keep up to date with research.: how many was this, from what background were participants and is this only based on one organisation? Participants in procurement roles were particularly and negatively vociferous about the influence of pharmaceutical representatives yet viewed the role of policing any promotional activity as a specialist nurse responsibility.: needs more detail When each organisation's expenditure on dressing types was discussed, community nurses and podiatrists were surprised to find that their use of antimicrobial dressings was higher than they had expected: how was this information presented – and how was it obtained?
--	---

	Participants did report that they adhered to national targets and regulations: what are these? Discussion: Podiatrists' utterances were coded as equally as other participants as we prompted all participants to respond to comments or questions if not spontaneously offered.: this needs to be in methods
--	--

REVIEWER	Dr Andrea Patey Centre for Implementation Research, Ottawa Hospital Research Institute, Canada
REVIEW RETURNED	29-Oct-2018

GENERAL COMMENTS	General Comments: The authors present a TDF based qualitative study with community health care providers investigating barriers and enablers to community wound care management. Through Focus Groups with over 50 HCPs the authors identified five relevant TDF domains that focussed on four main themes. Analysis of the focus group data is often quite labour intensive; conducting this type of work with a variety of HCP groups can make it more challenging. I commend the authors for doing this type of data collection and analysis. This study is relevant to those individuals interested in wound care management and I have no major concerns with the study. I do however, have some minor comments that the authors should to address. Minor Revisions:  1. In the Abstract, I think it would be helpful to report the domains that the authors identified as relevant in addition to the four themes. 2. The authors reported interviewing a number of groups of HCPs. Were the factors that influence wound care management the same across all groups interviewed or did they differ? Were the key themes identified, those common across all groups or were there themes specific to one HCP group interviewed. It would be helpful if this were more clearly articulated in the methods/results. 3. The authors described the behaviour as "managing wound care" whereby they discussed VLU diagnosis and use of compression bandages. I wonder if the authors thought about separating the behaviours since the barriers and enablers to one behaviour may be different than another. 4. It would also be helpful to report the findings from those domains that were not relevant and the reasoning for them not being relevant. Overall this is a well conducted investigation into the factors influencing wound care in Northern England. The authors have identified areas for improvement and potential targets for intervention. I look forward to seeing this study published.
--

VERSION 1 – AUTHOR RESPONSE

Comment by	Comment	Response	Changes to text highlighted by red (page and line numbers relate to the version with track changes)	Page/ Line
Editor	Please revise the title of your manuscript to include the research setting (country). This is the preferred format of the journal.	Revised	What factors influence community wound care in the UK? A focus group study using the Theoretical Domains Framework	Title page/1
Editor	Please revise the ‘Strengths and limitations’ section of your manuscript (after the abstract). This section should contain five short bullet points, no longer than one sentence each, that relate specifically to the methods.	A 5 th bullet point has been added and the remaining shortened. All now relate to the methods	 • This focus group study is the first to explore the factors that influence wound care and the reasons for known variation in practice. • Employing a qualitative methodology provided new insight into the role experiential learning and social influences play in determining clinical and procurement choices. • The focus group design stimulated discussion allowing participants to examine their own and others’ views and experiences. • The Theoretical Domains Framework provided a theoretical structure for developing a deeper understanding wound care. • The sample was taken from community healthcare organisations in the North of England, inclusion of participants from a larger geographical population may have provided different views. 	3/67-78
Editor	- Along with your revised manuscript, please include a copy of the COREQ checklist for reporting of qualitative research, indicating the page/line numbers of your manuscript where the relevant information can be found (https://academic.oup.com/in_tqhc/article/19/6/349/)	COREQ Checklist completed and uploaded as a separate file		

	179196 6/Consolidated- criteria-forreporting- qualitative)			
Editor	Please include a statement relating to obtaining informed	A statement has been added under Ethical considerations	Written informed consent was obtained from all participants.	9/204
	consent to participate in the study from all participants.			

Reviewer 1	Although the main research topic is of interest, there are several points that require clarification and more consistency in reporting across sections. The introduction is very brief and does not clearly situate the study within the authors' program of research and within related literature. The definition of complex wounds is not easy to understand by readers unfamiliar with wound care.	Thank you for reviewing our manuscript. The definition of complex wounds has been revised and the introduction has been expanded.	People with complex wounds (open wounds, such as foot, leg and pressure ulcers, burns, open trauma and surgical wounds that are difficult to heal),^{1,2} are more likely to be elderly and living with multimorbidity.³ Wound care normally begins with a comprehensive assessment of the person and their wound before implementation of appropriate interventions.⁴ Specific wound-related assessments include ankle brachial pressure index (ABPI) for people with venous leg ulcers. Wound treatment may involve wound cleansing followed by dressing use to manage exudate and protect the wound. Whilst dressings are used widely across wound types, with many different options available for use, there is currently no evidence that one dressing type is more clinically or cost effective than another, even in the case of relatively expensive anti-microbial dressings. In contrast there are effective first line treatments which should be widely used, such as the use of compression therapy for venous leg ulceration which is known to reduce time to wound healing.^{5,6}  1. Cullum N, Buckley H, Dumville J, et al. Wounds Research for Patient Benefit: a 5 year programme of research. Programme Grants Appl Res 2016;4(13) 2. Vowden K. Complex wound or complex patient? Strategies for treatment. Br J Community Nurs 2005;Suppl:S6, S8, S10 passim. [published Online First: 2005/06/10] 3. Gray TA, Dumville JC, Christie J, et al. Rapid research and implementation priority setting for wound care uncertainties. PLoS One 2017;12(12):e0188958. doi: 10.1371/journal.pone.0188958 [published Online First: 2017/12/06] 4. Coleman S, Nelson EA, Vowden P, et al. Development of a generic wound care assessment minimum data set. 	3/85-86 4/94-101
--	--	---	--------------------------------

			J Tissue Viability 2017;26(4):226-40. doi: 10.1016/j.jtv.2017.09.007 5. Ashby RL, Gabe R, Ali S, et al. VenUS IV (Venous leg Ulcer Study IV) - compression hosiery compared with compression bandaging in the treatment of venous leg ulcers: a randomised controlled trial, mixed-treatment comparison and decision-analytic model. Health Technol Assess 2014;18(57):1-293, v-vi. doi: 10.3310/hta18570 6. NICE. Chronic wounds: advanced wound dressings and antimicrobial dressings: Evidence summary [ESMPB2]. London: National Institute for Health and Care Excellence, 2016.	
--	--	--	---	--

	It would be helpful to clarify the role of nurses and specialist team members in wound care, and to clearly identify the population targeted in this qualitative study.	We have clarified the role of nurses and specialist teams and the target population.	In the UK, the management of people with complex wounds is mainly carried out in patients' homes or community clinics by community nurses with advice and support from specialist teams (nurses and medics with expertise in tissue viability, burns, vascular medicine or dermatology). Podiatrists also play a vital role in managing foot wounds, often working in conjunction with community nurses. People with complex wounds are more likely to be elderly, living with multimorbidity.	3/86-91 3/86
--	--	---	--	----------------------------

Reviewer 1	It is essential to explain what aspects or practices related to wound management were investigated in the previous study done by the authors, and the current study: was the focus on general wound care practices, on evidence-based guidelines, or just the use of equipment like the doppler? Once the practices of interest are identified, then it would be useful to provide more information on variation in its implementation as found in the previous study and to relate the TDF factors to these particular practices.	Both the survey and the focus group study focus on wound care in general. We have clarified this in the introduction. The paragraph added to the introduction on page 4 as above, explains what wound care involves. The survey revealed variation across clinical practice. We have added this. In this study, our aim was to surface factors that could potentially explain variations in wound care practice. We of course recognise that wound care practice is complex and multifaceted involving a wide range of individual tasks and activities. Given this, we recognise that any formal attempts to develop strategies to modify existing practices and behaviours will require a level of granularity beyond this study but does shed light on those TDF domains where future efforts should focus. We have added this statement to the limitations section	we conducted a survey to assess how healthcare professionals managed wound care across five community healthcare organisations in the North of England. The survey revealed unwarranted variation in clinical practice,	4/104106 4/107-108

Reviewer 1	Further, more information is needed on what is considered best practices and how the previous study findings	There is information in the discussion that refers to national guidelines that support best practice in wound care.	National guidelines exist to guide the use of specific products, ^{4 6-8} however, national standards to guide choice across the range of wound care products would reduce variation of product use and guide more rational prescribing. ⁹	22/543 545

	deviate from these best practices – to put the current qualitative study in context.	We have added information to the introduction to put the focus group study into context.	The findings are discussed in more detail elsewhere¹⁰ but in summary the survey revealed unwarranted variation in clinical practice, with general underuse of Doppler-aided measurement of ABPI,¹¹ underuse of compression therapy and ¹²). potential overuse of antimicrobial dressings.⁶ In the UK, variations in wound care are being recognised and addressed with initiatives such the Leading Change, Adding Value Nursing and Midwifery Framework^{13 14} however, there has been little formal exploration of drivers for this variation in the delivery of wound care and barriers to implementing the findings from current research evidence. In turn there is little intelligence to guide further research implementation and bring about meaningful practice change with the aim of maximising patient benefit. 4. Coleman S, Nelson EA, Vowden P, et al. Development of a generic wound care assessment minimum data set. J Tissue Viability 2017;26(4):226-40. doi: 10.1016/j.jtv.2017.09.007 5. Ashby RL, Gabe R, Ali S, et al. VenUS IV (Venous leg Ulcer Study IV) - compression hosiery compared with compression bandaging in the treatment of venous leg ulcers: a randomised controlled trial, mixed-treatment comparison and decision-analytic model. Health Technol Assess 2014;18(57):1-293, v-vi. doi: 10.3310/hta18570 6. NICE. Chronic wounds: advanced wound dressings and antimicrobial dressings: Evidence summary [ESMPB2]. London: National	4/107118
--	---	---	---	-----------------

			Institute for Health and Care Excellence, 2016. 7. NICE. Pressure ulcers: prevention and management Clinical guideline [CG179]: National Institute for Health and Care Excellence, 2014. 8. NICE. Diabetic foot problems: prevention and management [NG19]: National Institute for Health and Care Excellence, 2015. 9. Health Service Executive. National best practice and evidence based guidelines for wound management. Dublin, Ireland: Health Service Executive, 2009. 10. Gray TA, Rhodes S, Atkinson RA, et al. Opportunities for better value wound care: a multiservice, cross-sectional survey of complex wounds and their care in a UK community population. BMJ Open 2018;8(3):e019440. doi: 10.1136/bmjopen-2017-019440 [published Online First: 2018/03/25] 11. SIGN. Management of chronic venous leg ulcers: A national clinical guideline (120): Scottish Intercollegiate Guidelines Network, 2010. 12. Ashby RL, Gabe R, Ali S, et al. Clinical and cost-effectiveness of compression hosiery versus compression bandages in treatment of venous leg ulcers (Venous leg Ulcer Study IV, VenUS IV): a randomised controlled trial. Lancet 2014;383(9920):871-9. doi: 10.1016/S0140-6736(13)62368-5	
--	--	--	--	--

Reviewer 1	Is the aim of the study to examine wound care management and prescribing practices? Are these two different types of practices? How is each defined? The findings (related to factors influencing practices) presented do not allude to any specific practice.	The aim was to examine wound care in general. This has been clarified in the Abstract and in the Methods. For consistency, the term 'wound care' has now been used throughout.	This study explored the factors that influence community wound care to develop a deeper understanding of the current context of wound care and variation in practice. We conducted six focus group interviews to explore the factors that influence the delivery of wound care.	2/36-37 5/130131
Reviewer 1	The Methods section lacks explicit justification for the methodological decisions made. What is the target population for this qualitative study: only nurses and/or other members of the team responsible for wound management or for prescribing? Prescribing what: medical treatments or wound related materials?	All healthcare professionals (we have amended the manuscript throughout to use the term clinical professionals for consistency and to differentiate from participants in non-clinical roles (nonclinical professionals)) from participating provider organisations who cared for patients with complex wounds were eligible to participate in one of the multidisciplinary clinical professional focus group interviews. Non-clinical professionals from participating provider organisations who were responsible for the procurement of wound care products were eligible to participate in the non-clinical professional focus group interview. We explored the wound care role of clinical professionals as a whole which includes wound assessment, providing treatments such as cleansing, managing exudate and prescribing (or ordering) and applying products such as dressings, compression stockings and pressure relieving aids. This has been clarified in the Introduction (as	Eligibility included community-based clinical professionals who cared for patients with complex wounds or non-clinical professionals who were involved in the procurement of wound care products. Clinical professionals included community nurses, podiatrists, tissue viability or burns specialist nurses, wound research nurses and clinical nurse managers (who had a clinical role, managed a team of community nurses and were responsible for wound product procurement decisions). Non-clinical professionals included: medicines optimisation pharmacists, procurement leads, procurement advisors and medicines management leads.	7/142-148

		mentioned above) and the Methods		
	Were participants in this qualitative study selected from those who took part in	Both studies are part of a NIHR CLARHC GM programme of wound care research which involves working collaboratively with partner healthcare provider organisations	No change to manuscript warranted	
	the previous quantitative study?	to improve wound care. Participants were drawn from the same target populations. It is possible that participants from this study took part in the survey as this targeted all wound care services but as the survey questionnaires were anonymised we would not know for sure whether they were involved.		

Reviewer 1	What were the community settings targeted for the study: primary care, community care, ambulatory care or home care?	In the UK, community service settings include patients' homes and community clinics. This has been clarified in the introduction.	In the UK, the management of people with complex wounds is mainly carried out in patients' homes or community clinics by community nurses	3/86-91
Reviewer 1	What were the different professional groups included in the group interviews? Should the results then be presented by professional groups instead of being lumped regardless of professional groups? Or, where the emerging themes the same across professional groups? This point should be clarified in light of the limitation stated that one podiatrist participated in the study.	Professional groups included clinical professionals who cared for patients with complex wounds: specialist nurses, community nurses, a research nurses, clinical managers, podiatrists and nonclinical professionals who were responsible for the procurement of wound care products as presented in Table 2. Many of the themes relate to participants across the study; where findings are specific to a particular professional group we have added this to clarify. Seven podiatrists participated in the study as presented in Table 2	Clinical professionals across all groups expressed feeling the pressure of increased workloads. Community nurses reported that specialist clinics were being cut, Community nurses and podiatrists voiced concern that undue time was spent gathering required patient information due to poor referral information supplied by hospital staff The majority of clinical professionals reported that specialist leg ulcer clinics had been cut resulting in a greater number of home visits for community nurses. Only specialist nurses, had attended a university-based postregistration wound care course All clinical professionals viewed wound care knowledge amongst hospital, general practitioners and nursing home-based nurses to be poor, Clinical professionals reported that team support alleviated some of the current workload pressures Non-clinical professionals and specialist nurses were aware of the high expenditure	12/262 12/272 12/280282 13/290292 15/351353 15/355357 17/400402 18/437-438

Reviewer 1	Why were clinical nurse managers included? Do they provide direct patient care?	Community Clinical managers do have a clinical role. They are very experienced and manage and support a team of community nurses in making wound care decisions as well as service decisions. They are also involved in wound care product procurement decisions and therefore important to include. Their role has been clarified.	Clinical professionals included community nurses, podiatrists, tissue viability or burns specialist nurses, wound research nurses and clinical nurse managers (who had a clinical role, managed a team of community nurses and were responsible for wound product procurement decisions).	7/144-146
Reviewer 1	Please, provide explicit recommendations for sample size in qualitative research; this is important to determine the adequacy of the accrued sample size in total and for each professional group.	Few empirical studies exist to guide researchers in determining the number of focus groups necessary. ^{15 16} Many researchers cite theoretical saturation as a method for determining nonprobability sample sizes in qualitative research. ¹⁶ We decided a priori to include all CLAHRC GM partner provider organisations and hold a focus group interview for each organisation. Although we anticipated that we would reach data saturation in 3-4 focus group interviews, we planned to hold 6 focus groups (50-60 participants in total) to incorporate all provider organisations and a range professionals. The authors are not aware of any clear guidelines to determine the optimum size of a focus group. Traditionally, 10-12 participants have been acceptable, however some qualitative researchers believe than more than 10 participants are difficult	There were five multidisciplinary focus group interviews for clinical professionals; one for each participating provider organisation. Four were drawn from provider organisations in one defined geographical area with a fifth conducted in a different geographical area but similar urban conurbation in the North of England; chosen for its well-established links with university researchers as a comparison to the other organisations where collaborative partnerships with university researchers were in their infancy. A separate focus group interview was held for non-clinical professionals. As the themes for clinical and non-clinical focus group interviews differed, we chose this format to maintain focus and create an optimum	7/149154 7/158160 8/167-171

		to control and limit each person's opportunity to share insights. Small focus groups with four to six participants are becoming increasingly popular.^{17 18} We decided that 8-10 would be sufficient to accommodate a range of professionals in each group but not too large to limit the opportunity to share insights. References have been added to the Methods.	environment for free flowing discussions. As participants were drawn from a relatively homogeneous population and the interview schedules were focused on specific aspects of wound care and wound product procurement, we anticipated that we would reach data saturation within three to four focus group interviews, however, we aimed to recruit 50-60 participants in total across the six groups (8 to 10 participants per group), based on recommendations from existing literature¹⁵⁻¹⁸ to incorporate all partner provider organisations using the format described above.	
--	--	--	--	--

		15. Carlsen B, Glenton C. What about N? A methodological study of sample-size reporting in focus group studies. BMC Med Res Methodol 2011;11:26. doi: 10.1186/1471-2288-11-26 [published Online First: 2011/03/15] 16. Guest G, Namey E, McKenna K. How Many Focus Groups Are Enough? Building an Evidence Base for Nonprobability Sample Sizes. SAGE		
--	--	---	--	--

		journals 2016;29(1):3-22. doi: https://doi.org/10.1177/1525822X16639015 17. Krueger RA, Casey MA. Chapter 4. Participants in a Focus Group. Focus Groups: A Practical Guide for Applied Research. 5 ed. Thousand Oaks, C.A.: Sage Publications Inc 2015:79101. 18. Tang KC, Davis A. Critical factors in the determination of focus group size. Fam Pract 1995;12(4):474-5. [published Online First: 1995/12/01]		
Reviewer 1	Page 6, line 45-46 – the data do not “summarize” but “describe characteristics” of participants. Please be careful with terminology.	This has been amended.	these data were used to describe the participants involved	8/179
Reviewer 1	What specific barriers were discussed?	We discussed the factors that influence wound care and prompted participants to consider enablers and constraints (barriers) such as a formulary, how restrictive is was, was too much choice an enabler or a constraint? What effect did product cost and availability have on choice? How does knowledge effect wound care, were there adequate training opportunities? Is research evidence used? Is there time to look for research? How research disseminated within the organisation? More detail is presented in Appendix 1		

Reviewer 1	What strategies were used to maintain reliability and validity in this qualitative study?	As per recommended strategies by Noble and Smith 2015¹⁹ for ensuring credibility in qualitative research, strategies included: piloting of interview schedules, independent coding by two researchers, cross checking against recordings for accuracy, respondent validation, reporting verbatim to support findings and acknowledging sampling bias. A COREQ checklist has also been submitted 19. Noble H, Smith J. Issues of validity and reliability in qualitative research. Evid Based Nurs 2015;18(2):34-5. doi: 10.1136/eb-2015102054 [published Online First: 2015/02/06]	Interview schedules were piloted by specialist nurses, clinical managers and a procurement lead, after which minor amendments were made.  1. Transcripts were then coded inductively by TG to establish the main themes. 2. An analytical framework was developed by two researchers (TG and PW). The thematic codes generated inductively were aligned with the 14 theoretical domains of the TDF. Respondents validated the accuracy and completeness of the findings¹⁹ following a verbal summary (taken from the field notes) at the end of each focus group interview and a post-analysis report sent via email. The main limitation is the sample which was taken from community healthcare provider organisations in the North of England and included only one research-active organisation.	8/193 Figure 1 8/194197 24/566-567
Reviewer 1	The findings section could be improved. It would be helpful to summarize the main characteristics of participants in the text, rather than just referring to the table.	Participant characteristics have been added to the text.	Participants comprised 46 clinical professionals (ten specialist nurses (19%), 25 community nurses (46%), seven podiatrists (13%), three clinical managers (5%), one research nurse (2%)) and eight non-clinical professionals (15%). Wound care experience was extensive (mean 14.6 years, SD 8.8) amongst healthcare professionals (Table 2).	10/242-245

Reviewer 1	It is a bit difficult to follow / understand the findings or themes that emerged: It is unclear what specific practices / behaviors were discussed, and were found to be influenced by the factors listed. It is unclear if the findings / identified factors were mentioned by all professional groups across all organizations, or if some factors were more relevant to some groups in some organizations, which is	The introduction to the findings has been amended to clarify the salient TDF domains. We have amended the findings throughout to more clearly define the TDF themes and the sub-themes that emerged through relevance to variation in wound care, wound care services the implementation of research findings. As per response to previous comment above, we have amended the findings to specify which professional group specific findings refer to where appropriate. Where 'participants' are mentioned, this indicates	Five TDF domains dominated: Environmental context and resources, Knowledge, Skills, Social influences and Behaviour regulation . The domains of knowledge and skills were closely linked and frequently overlapped, therefore, we combined these. We did not code any source data to the domains of Emotion and Intentions and found the remaining six domains to overlap with the five dominant domains. We have therefore, focused on the five key domains which best explain the variation in wound care and services and the underuse of research evidence	11/251256
------------	--	---	---	-----------

	expected under different environmental contexts.	that the finding relates to participants across the groups.		
Reviewer 1	The quotes presented are not necessarily consistent with or do not clearly illustrate the respective theme. For some themes, no quotes are presented.	We have revised the manuscript to align the findings to our aim of exploring the factors that influence wound care to explain the variations in practice and underuse of research found in the survey. We have removed some of the sub-themes that are less relevant and we have added quotes to support the findings. We hope that this has improved consistency.	"But then at the same time they'd spy the competition and they'd basically suggest that their products are equivalent to those products that were already on the shelfand then we were inundated with requests for new products." (Non-clinical professional) "I can't know everything about all dressings, and therefore you often stick to what you know and you don't often have time to look at research." (Non-clinical professional) "And as healthcare professionals it's not built into our contracts to do research...there's no time put aside." (Specialist nurse) "You can police this a little bit more in acute, can't you, but in the community we	16/369371 17/390391 17/393394 18/416417 18/426-427

			were fighting a losing battle with the reps when they're just given free range to provide training."(Non-clinical professional) "She (the patient) read that honey was good and she thought I'll go and buy my own.... and swore it did the trick, so who are we to argue with her?" (Community nurse)	
Reviewer 1	There is no explanation of how exactly the identified factors influence what particular practice. It is not always clear what each factor is exactly about and codes reflect it. The authors may want to consider replacing the table listing the questions asked to explore each theoretical domain by a table that summarizes the domains, factors that emerged from the qualitative data analysis and related to each domain, the specific codes reflecting the factors, and the most illustrative quote.	We have revised Table 1 adding a column with examples of wound care related behaviours to make it clearer to the reader the link between the key themes identified and within relevant TDF domains. We have added a coding tree to help illustrate the findings. We believe this upfront presentation now adds sufficient context and clarity for the results section that follows.	Table 1: The Theoretical Domains Framework: domains, definitions and examples of behaviours related to wound care and wound product procurement Figure 2. Coding Tree showing the four salient domains with connected sub themes	6/139 Figure 2 5/614

Reviewer 1	The discussion appears to be selective, addressing specific factors (some of which were mentioned very briefly in the results section, such as the issue of technology) instead of the identified domains – this gives the impression of inconsistency across sections.	The discussion has been amended to align with the findings and the less relevant sections have been removed e.g. the issues of technology, to aid flow and clarity.		
Reviewer 2	The subject matter is important and interesting. The paper is well written, informative and reads well. There are some issues, however, that impact on generalisability. The study was carried out in only one geographic area and with only one research-active organisation. I think the authors could have discussed this in a little more detail in the discussion / limitations and justified why this methodology will not limit generalisability within the UK in their view. This is the main limitation of the study.	Thank you for reviewing our manuscript and for your comments regarding the subject and the writing. We have moved this limitation to the top and identified it as the main limitation. We have discussed the limitations in more detail and incorporated the reasons why we were only able to recruit one researchactive organisation.	The main limitation is the sample which was taken from community healthcare provider organisations in the North of England and included only one research-active organisation. Inclusion of participants from a larger geographical population may have provided different views, however, we captured many of the issues affecting healthcare (such as work pressures, staff shortages and limited resources) across the UK ²⁰⁻²² and further afield ^{23 24} due to the financial healthcare crisis worldwide. We would have preferred to include more than one research-active organisation but due to the limited number of research-active organisations within our geographical area as well as funding and time limitations we could not recruit more. 20. Robertson R, Wenzel L, Thompson J, et al. Understanding NHS financial pressures How are they affecting patient care? London: The King's Fund, 2017. 21. Maybin J, Charles A, Honeyman m. Understanding quality in district nursing services: Learning from patients,	24/566-572

			carers and staff. London: The King's Fund, 2016. 22. Jackson C. Implementing a District and Community Nursing workload tool, to determine safe staffing levels and skill mix in a community care provider organisation: An economic assessment of potential benefits for workforce planning. Kent: Canterbury Christ Church University 2016. 23. Saltman RB. The impact of slow economic growth on health sector reform: a cross-national perspective. Health Econ Policy Law 2018:1-24. doi: 10.1017/S1744133117000445 24. Karanikolos M, Mladovsky P, Cylus J, et al. Financial crisis, austerity, and health in Europe. Lancet 2013;381(9874):1323-31. doi: 10.1016/S01406736(13)60102-6	
--	--	--	---	--

Reviewer 2	The introduction needs more detail in my opinion on this study rather than a previous study, including detail of wound care pathways and evidence for existence of variation in practice ie the basis for why this study is needed.	The introduction has been expanded to include more information about the area of study including a description of what wound care involves and the evidence for variation.	Wound care normally begins with a comprehensive assessment of the person and their wound before implementation of appropriate interventions.⁴ Specific wound-related assessments include ankle brachial pressure index (ABPI) for people with venous leg ulcers. Wound treatment may involve wound cleansing followed by dressing to manage exudate and protect the wound. Whilst dressings are used widely across wound types, with many different options available for use, there is currently no evidence that one dressing type is more clinically or cost effective than another, even in the case of relatively expensive anti-microbial dressings. In contrast there are effective first line treatments which should be widely used, such as the use of compression therapy for venous leg ulceration which is known to reduce time to wound healing^{5 6} In the UK, variations in wound care are being recognised and addressed with initiatives such the Leading Change, Adding Value Nursing and Midwifery Framework^{13 14} however, there has been little formal exploration of drivers for this variation in the delivery of wound care and barriers to implementing the findings from current research evidence. In turn there is little intelligence to guide further research implementation and bring about meaningful practice change with the aim of maximising patient benefit.	4/94-101 4/113-118
--	---	---	----------------------------------

			4. Coleman S, Nelson EA, Vowden P, et al. Development of a generic wound care assessment minimum data set. J Tissue Viability 2017;26(4):226-40. doi: 10.1016/j.jtv.2017.09.007 5. Ashby RL, Gabe R, Ali S, et al. VenUS IV (Venous leg Ulcer Study IV) - compression hosiery compared with compression bandaging in the treatment of venous leg ulcers: a randomised controlled trial, mixed-treatment comparison and decision-analytic model. Health Technol Assess 2014;18(57):1-293, v-vi. doi: 10.3310/hta18570 6. NICE. Chronic wounds: advanced wound dressings and antimicrobial dressings: Evidence summary [ESMPB2]. London: National Institute for Health and Care Excellence, 2016. 13. Adderley U, Evans K, Coleman S. Reducing unwarranted variation in chronic wound care. Wounds UK 2017;13(4):22-27. 14. England N. Leading Change, Adding Value: A framework for nursing, Midwifery and care staff London, UK: NHS England; 2016 [Available from: https://www.england.nhs.uk/wp-content/uploads/2016/05/nursing-framework.pdf]	
--	--	--	--	--

Reviewer 2	The themes reported in the results are a mix of information from participants and facts. It is sometimes unclear which was which. Many of the participant views are not backed up by facts and it is unclear if these are unchecked views or views based on checked facts?	All the information detailed in the findings has come from participants. The findings have been revised throughout to clarify who has provided the information. The number of verbatim quotes have been added to support the findings.	Clinical professionals across all groups expressed feeling the pressure of increased workloads. Community nurses reported that specialist clinics were being cut, Community nurses and podiatrists voiced concern that undue time was spent gathering required patient information due to poor referral information supplied by hospital staff The majority of clinical professionals reported that specialist leg ulcer clinics had been cut resulting in a greater number of home visits for community nurses. Only specialist nurses, had attended a university-based postregistration wound care course All clinical professionals viewed wound care knowledge amongst hospital, general practitioners and nursing home-based nurses to be poor, Clinical professionals reported that team support alleviated some of the current workload pressures Non-clinical professionals and specialist nurses were aware of the high expenditure “But then at the same time they’d spy the competition and they’d basically suggest that their products are equivalent to those products that were already on the shelfand then we were inundated with requests for new	12/262 12/272 12/280282 13/290292 15/351353 15/355357 17/400402 18/437438 16/369371 17/390391 17/393394
------------	---	---	---	--

			products.” (Non-clinical professional) “I can’t know everything about all dressings, and therefore you often stick to what you know and you don’t often have time to look at research.” (Non-clinical professional) “And as healthcare professionals it’s not built into our contracts to do research...there’s no time put aside.” (Specialist nurse)	
--	--	--	---	--

			“You can police this a little bit more in acute, can’t you, but in the community we were fighting a losing battle with the reps when they’re just given free range to provide training.”(Non-clinical professional) ”She (the patient) read that honey was good and she thought I’ll go and buy my own.... and swore it did the	17/416417 18/426-427
--	--	--	---	------------------------------------

			trick, so who are we to argue with her?" (Community nurse)	
Reviewer 2	Abstract: ability of provide holistic care: typo Organisations need to develop strategies to apportion resources wisely and redeploy skills: I do not think this can be concluded from this study and is not in the main conclusion?	The sentence referring to holistic care has been revised. 'Organisations need to develop strategies to apportion resources wisely and redeploy skills' has been removed from the abstract.	Workforce pressures and limited resources are perceived to impede care by reducing patient access to services and the ability to provide holistic care	2/55-56
	Introduction:open trauma and surgical wounds: what is the definition of a complex wound – these seem to be wounds that should heal simply? Are burns included? Needs more detail on wound care pathways and also evidence for variation in practice and need for this study?	The definition refers to open wounds that are difficult to heal. Burns have been added and the sentence and it has been revised. As above, the Introduction has been extended to include more detail about wound care pathways and the evidence for variation.	People with complex wounds (open wounds, such as foot, leg and pressure ulcers, burns, open trauma and surgical wounds that are difficult to heal), ^{1,2}	3/85-86
Reviewer 2	Methods: Purposive sampling was used to ensure that we recruited participants with relevant community clinical, management and procurement experience.: more detail please ie what is relevant?	Relevant experience included clinical experience of caring for patients with complex wounds or managing the procurement of wound products in a non-clinical role. The eligibility criteria has been moved to the front of this section to explain this and the added information in the introduction describing what wound care involves provides further detail of the wound care role of clinical professionals.	Eligibility included community-based clinical professionals who cared for patients with complex wounds or non-clinical professionals who were involved in the procurement of wound care products. Clinical professionals included community nurses, podiatrists, tissue viability or burns specialist nurses, wound research nurses and clinical nurse managers (who had a clinical role, managed a team of community nurses and were responsible for wound product procurement decisions). Non-clinical professionals included: medicines optimisation pharmacists, procurement	7/142-148

			leads, procurement advisors and medicines management leads.	
--	--	--	---	--

Reviewer 2	The focus group interviews were arranged by professional group: No multidisciplinary groups – why was this decision maderegularly cared for patients with complex wounds: what does regularly mean?	All groups were multidisciplinary. The wording has been amended to clarify this. Clinical and non-clinical professionals were separated to maintain focus. The clinical professional groups' focus was related to wound care and the non-clinical professional group was related to wound product procurement. This has been explained in the Introduction and the Methods.	We wanted to better understand the current context of community wound care and how research evidence informs care delivery. We were keen to explore clinical viewpoints and also, because of possible factors linked to the availability and use of wound products with a limited evidence-base, we involved non-clinical staff responsible for procurement processes.	5/123126
			There were five multidisciplinary focus group interviews for clinical professionals;	7/149
			Clinical professionals included community nurses, podiatrists, tissue viability or burns specialist nurses, wound research nurses and clinical nurse managers (who had a clinical role, managed a team of community nurses and were responsible for wound product procurement decisions)	7/144146
			A separate focus group interview was held for non-clinical professionals. As the themes for clinical and non-clinical focus group interviews differed, we chose this format to maintain focus and create an optimum environment for free flowing discussions.	7/158138
			Non-clinical professionals included: pharmacists, procurement leads,	7/146-148

			procurement advisors and medicines management technicians.	
Reviewer 2	The NIHR CLAHRC Greater Manchester Wounds Research PPI Forum provided views on their experiences with healthcare professionals and wound care services.: this is unclear – how were patients and public involved?	This has been revised to clarify PPI Involvement	Patients and the public were not involved in the interviews, however, views expressed by members of the NIHR CLAHRC Greater Manchester Wounds Research PPI Forum regarding their experiences with healthcare professionals and wound care services were used to inform some of the questions and prompts for the focus group discussions.	9/207-210
Reviewer 2	Specialist nurses, clinical managers and a procurement lead were involved in piloting the interview schedules, after which minor amendments were made. Member validation was performed at the end of each focus group	Thank you for noting this, this section should not have been placed under PPI. It has been moved to the Methods. We have altered the wording to describe how we invited participants to validate the findings. None of the participants requested any changes to the transcripts or interpretation of findings they were happy	Interview schedules were piloted by specialist nurses, clinical managers and a procurement lead, after which minor amendments were made. Respondents validated the accuracy and completeness of the findings ¹⁹ following a verbal summary (taken from the field notes) at the end of each interview and a post-analysis report sent via email	9/192193 9/193-196

	interview and following analysis: how does this relate to PPI and what is member validation please? No corrections were requested.: why was this?	that they were accurate and complete. The revised wording reflects this.		
Reviewer 2	Results: The authors state purposive sampling was undertaken – but also state that 87% of participants were female participants and only one research nurse?? I think this requires more explanation of justification.	Purposive sampling was carried out to ensure clinical and/or procurement experience, gender was not relevant. As approximately 80% of NHS Agenda for Change staff²⁵ are female. We therefore, feel that the finding of 87% of participants in our study were female (with 72% nurses) is not surprising. The participating research-active organisation was the only organisation to employ wound research nurses to work closely with clinical teams, therefore, it would have been difficult to recruit more. An explanation has been added to the limitations. 25. NHS Employers. Gender in the NHS 2018 [Available from: https://www.nhsemployers.org//media/Employers/Images/2018-D-and-infographics/Gender-in-the-NHS-2018.pdf.	Purposive sampling was used to ensure that we recruited participants with relevant clinical and/or procurement experience Only one research nurse was recruited but as the research-active organisation was the only organisation to employ wound research nurses it is not surprising that we could only recruit one.	7/141142 24/579-581
Reviewer 2	As referred to above, the pharmaceutical industry is also influential; providing training and education around product use, although this varied across organisations depending on the capacity of specialist nurses to limit access and monitor sessions: where is this information from?	This has been amended to clarify that participants voiced this concern	As referred to above, all participants were concerned about the influence of the pharmaceutical industry it was felt that this influence varied depending on the capacity of specialist nurses to limit access and monitor training sessions.	17/410-413

Reviewer 2	In order to standardise and better regulate prescribing behaviour across several provider organisations including primary care, plans were in progress to produce a regional formulary: where is this information from? For this reason, only obligatory monitoring appeared to be conducted.: fact or comment?	As mentioned earlier, we have revised the manuscript to align the findings to our aim of exploring the factors that influence wound care to explain the variations in practice and underuse of research found in the survey. We have removed this and other sub-themes that are less relevant. We hope that this has improved clarity and consistency.		
Reviewer 2	Participants across all focus group interviews expressed feeling the pressure of increased workloads and Another concern that caused unnecessary stress and treatment delays was the poor referral information received from hospital medical and nursing staff for patients discharged to the care of community nurses (p11): was there any evidence for this and other findings from different sources?	Community nurses and podiatrists voiced concern regarding poor referral information. This has been revised to clarify.	Community nurses and podiatrists voiced concern that undue time was spent gathering required patient information due to poor referral information supplied by hospital staff.	13/280-282
Reviewer 2	Three community services had amalgamated with hospital (acute) services in the last five years and this was reported to have had a profound effect on staff morale.: what effect?	This has been revised to make it more succinct.	Participants from organisations that managed both hospital (acute) and community services felt that resourcing prioritised the acute service at the expense of the community service.	14/304-305
Reviewer 2	In terms of training for postqualified nurses, universitybased wound care courses	All findings were reported by participants but we have revised this section to make it more succinct.	Only specialist nurses had attended a university-based postregistration wound care course.	15/351352

	were available but these were mainly accessed by nurses preparing for a specialist wound care role and were not often attended by generalist nurses, resulting in a limited number of nursing staff with a higher level of wound care knowledge. Participants viewed wound care knowledge across other services to be particularly poor especially amongst hospital and nursing homebased nurses and GPs.: is the first part fact?		All clinical professionals viewed wound care knowledge across other services (hospital, primary care and nursing homes) to be poor, which increased their workload if aspects of care, documentation or prescription information were incomplete.	15/355-358
Reviewer 2	Only those participants from the healthcare organisation with a history of collaborative wound care research indicated that they actively sought to keep up to date with research.: how many was this, from what background were participants and is this only based on one organisation?	Yes this was is a multidisciplinary group of clinical professionals from one organisation as per the other four multidisciplinary clinical professional groups. The methods and findings has been revised in several places to clarify.	There were five multidisciplinary focus group interviews for clinical professionals; one for each participating provider organisation. Four were drawn from provider organisations in one defined geographical area with a fifth conducted in a different geographical area but similar urban conurbation in the North of England; chosen for its wellestablished links with university researchers as a comparison to the other organisations where collaborative partnerships with university researchers were in their infancy. Only participants from the provider organisation with a history of collaborative wound care research indicated that they actively sought to keep up to date with research. Specialist	7/149154 16/377-380

			nurses from this focus group talked about their established links with university researchers and their involvement in co-producing wounds research with academics.	
Reviewer 2	Participants in procurement roles were particularly and negatively vociferous about the influence of pharmaceutical representatives yet viewed the role of policing any promotional activity as a	Participants in procurement roles were from the non-clinical professional group. A quote has been added to provide more detail.	Non-clinical pParticipants in procurement roles were particularly and negatively vociferous about the influence of pharmaceutical representatives yet viewed the role of policing any promotional activity as a specialist nurse responsibility. "You can police this a little bit more in acute, can't you, but in the community we were fighting a losing battle with the reps when they're just given free range to provide training." (Non-clinical professional)	17/413415 18/416-417

	specialist nurse responsibility.: needs more detail			
Reviewer 2	When each organisation's expenditure on dressing types was discussed, community nurses and podiatrists were surprised to find that their use of antimicrobial dressings was higher than they had expected: how was this information presented – and how was it obtained?	In collaboration with partner procurement teams, we collected dispensing data via ePACT (an application which allows users to electronically access prescription data) for Greater Manchester and presented it via power point presentations to each group. We report in the Methods the discussion is linked to procurement data. We have revised the paragraph in the Findings to make it more succinct. As we are over the word limit we have removed	The discussion explored specific behaviours linked to of the TDF domains and reactions to site-specific, regional and national procurement data using the questions and prompts outlined in Appendix 1. Community nurses reported that antimicrobial dressings (particularly silver-impregnated dressings) were used for individual patients for a two-week trial period and then reviewed, however, they acknowledged that if use	8/184185 18/434-439

		the first part of the paragraph as the overuse of antimicrobial dressings can be explained without adding this extra information.	was not closely monitored there was potential for misuse. Non-clinical professionals and specialist nurses were aware of the high expenditure on antimicrobial dressings but acknowledged difficulties in monitoring effectively and providing adequate training and support due to capacity issues.	
Reviewer 2	Participants did report that they adhered to national targets and regulations: what are these?	Again this section is not directly related to the variation or underuse of research evidence, therefore, we have removed it.		
Reviewer 2	Discussion: Podiatrists' utterances were coded as equally as other participants as we prompted all participants to respond to comments or questions if not spontaneously offered.: this needs to be in methods	We have amended the limitation to clarify that podiatrists were as vocal as other clinical professionals and we have transferred the part relating to prompting to the Methods as suggested.	Podiatrists' utterances were coded as equally as other professional groups, therefore, we feel that podiatrists' views have been incorporated adequately. We continued to prompt if responses were not spontaneously offered to encourage full participant engagement.	24/577579 8/190-191
Reviewer 3	In the Abstract, I think it would be helpful to report the domains that the authors identified as relevant in addition to the four themes.	This has been added to the results section of the abstract.	We found the TDF domains: Environmental context and resources, Knowledge, Skills, Social influences and Behaviour regulation to best explain the variation in wound care and services and the underuse of research evidence.	2/46-47
	The authors reported interviewing a number of groups of HCPs. Were the factors that influence wound	Thank you, this has been consistently highlighted by all reviewers. We have therefore, revised the findings throughout	Clinical professionals across all groups expressed feeling the pressure of increased workloads. Community nurses reported that specialist clinics were being cut,	12/262 12/272

	care management the same across all groups interviewed or did they differ? Were the key themes identified, those common across all groups or were there themes specific to one HCP group interviewed. It would be helpful if this were more clearly articulated in the methods/results.	to clarify who has provided the information.	Community nurses and podiatrists voiced concern that undue time was spent gathering required patient information due to poor referral information supplied by hospital staff The majority of clinical professionals reported that specialist leg ulcer clinics had been cut resulting in a greater number of home visits for community nurses. Only specialist nurses, had attended a university-based postregistration wound care course All clinical professionals viewed wound care knowledge amongst hospital, general practitioners and nursing home-based nurses to be poor, Clinical professionals reported that team support alleviated some of the current workload pressures Non-clinical professionals and specialist nurses were aware of the high expenditure	12/280282 13/290292 15/351353 15/355357 17/400402 18/437-438
	The authors described the behaviour as “managing wound care” whereby they discussed VLU diagnosis and use of compression bandages. I wonder if the authors thought about separating the behaviours since the barriers and enablers to one behaviour may be different than another.	In this study, our aim has been to surface factors that could potentially explain variations in wound care practice. We of course recognise that wound care practice is complex and multifaceted involving a wide range of individual behaviours. Given this, we recognise that any formal attempts to develop strategies to modify behaviours will require a level of granularity beyond what is available in the data presented. Our study does shed light on those domains where future efforts should focus. This has been added as limitation to the discussion	Finally, our aim in this study has been to surface factors that could potentially explain variations in the delivery of wound care. We of course recognise that wound care practice is complex and multifaceted involving a wide range of individual behaviours. Given this, we recognise that any formal attempts to develop strategies to modify existing behaviours will require a level of granularity beyond what is available in the data presented. Our study does shed light on those domains where those future efforts should focus.	25/595-599

Reviewer 3	It would also be helpful to report the findings from those domains that were not relevant and the reasoning for them not being relevant.	We have added a sentence to acknowledge the domains that we did not directly include in the findings and the reason why we focused on the five salient domains.	We did not code any source data to the domains of Emotion and Intentions and found the remaining six domains to overlap with the five dominant domains. We have therefore, focused on the five key domains which best explain the variation in wound care and the underuse of research evidence.	12/253-256
		We are over the word limit and therefore have not been able to add any further information about the less salient domains.		
Reviewer 3	Overall this is a well conducted investigation into the factors influencing wound care in Northern England. The authors have identified areas for improvement and potential targets for intervention. I look forward to seeing this study published.	Thank you for reviewing our manuscript and for you positive comments.		

VERSION 2 – REVIEW

REVIEWER	Amber Young University Hospitals Bristol NHS Trust, University of Bristol
REVIEW RETURNED	15-Mar-2019

GENERAL COMMENTS	In general: As discussed in my previous review of this work, the subject matter is important and interesting. The paper is well written. There are still some issues that in my opinion need to be addressed however:  • The study aim needs to be clearer with respect to: complex vs non-complex wounds, definition of wound care and links to academic support • The authors need to address/discuss bias in terms of participant choice in the discussion / limitations • Was the conclusion regarding the importance of academic support based on only one research nurse and one organisation? • Please kindly defend the lack of PPI input. • Data analysis methods are unclear especially in regards to comparing and collating data from the two different groups. There
---

	is little information on qualitative analysis (I cannot find Figure 1 which may explain this?)  • Is there evidence to support the staff views on resources, time available etc or are these staff perceptions? • As this is based on community wound care – what was the role of the specialist nurses in the focus groups? • Discussion (lines 483-490) – I am not clear if this is derived from the study findings or is a conclusion? • The discussion could be somewhat shortened in my opinion with respect to the study findings but also needs more links to previous published work.. Specific comments: Abstract: Conclusion:  • what does: 'are perceived' mean? By whom? • The link between university and health care organisations is not in the methods and therefore shouldn't be in the conclusion? Is this based on one participant? Introduction:  • comprehensive assessment of the person: please detail what this means • Is it worth adding detail of the aims of wound dressings in the introduction – ie what does clinically effective mean in this context and how is this assessed – healing time, pain, infection, exudate?? • Line 100: do you mean that there is no evidence for dressing type but good evidence for compression therapy? Are there any other treatments with evidence? • Line 110 – just compression therapy?? • Line 121: dressings or all wound care? • Is this for complex wounds or all wounds? • Is there an aim to assess the link to University support? This is mentioned in the methods? • I think the last sentence should be in Methods and leave the aim as the last intro sentence Methods:  • Line 130: as above – complex or all wounds? • Line 159: why did they differ and in what way? Which format is referred to in line 159? • Line 167-171 seems to contradict itself? • Please add details on assessing the link to University support, defining complex wounds for participants, defining wound care • Data collection: how did the questions vary between the two groups and how were the data from the two groups collated? How was the data formally linked to the TDF domains? • What were the themes for the interviews – was it completely open at the start – eg was photography always asked about, time for dressing / dwound care?? Was the topic guide modified for subsequent interviews? • Please justify why patients were not involved in any way – could they not have been involved to assess questions asked? • Data analysis – how were the themes extracted and how were the two groups compared? Was a software package used? • I cannot find Figure – apologies iof I have missed it? Results  • Why only one research nurse? • Is there evidence for understaffing (line 274) or is this a perception? • A similar question regarding line 281?
--	---

	 • Lines 411-413: I am not clear what this means? • Line 437: what does 'mis-use' mean? Discussion:  • Line 465: for complex wounds alone? • Lines 483-490: is this a conclusion or does this come from the results? • Was the 'restrictive formulary' based on evidence? • Lines 579-581: it may not be surprising but what bias would this confer to the findings
--	--

REVIEWER	Andrea Patey Centre for Practice-Changing Research, ttawa Hospital Research Institute, Canada
REVIEW RETURNED	02-Apr-2019

GENERAL COMMENTS	I believe the authors have adequately responded to concerns based by myself and the other reviewers.
--

VERSION 2 – AUTHOR RESPONSE

Comment by	Comment	Response	Changes to text highlighted by red (page and line numbers relate to the version with track changes)	Starting Page/ Line
Reviewer 2: Amber Young	The study aim needs to be clearer with respect to: complex vs non-complex wounds, definition of wound care and links to academic support	We have clarified the aim to specify that the study focused on complex wounds only. The definition of wound care has been reworded to specify what is involved	Our aim was to identify and explore factors that influence care in community settings for people with complex wounds. Care of complex wounds in community settings normally includes a comprehensive assessment of the person and their wound, (involving demographics, risk factors for wound healing, quality of life measures, wound status, wound parameters and symptoms) specific wound-related assessments such as ankle brachial pressure index (ABPI) for people with venous leg ulcers and implementation of appropriate interventions.¹ Interventions may involve wound cleansing followed by dressing to manage exudate and protect the wound.	4/108 3/84

Reviewer 2	The authors need to address/discuss bias in terms of participant choice in the discussion / limitations	In the original and revised submission we highlighted the limitation of the geographical area and the limited availability of research active organisations. With regards to participant choice, our inclusion criteria did include the full range of healthcare professionals involved in community wound care and non-clinical professionals involved in the procurement of wound products.	The main limitation is the sample which was taken from community healthcare provider organisations in the North of England and included only one research-active organisation. Purposive sampling was used to ensure that we recruited participants with relevant clinical and/or procurement experience. Eligibility included community-based clinical professionals who cared for patients with complex wounds or non-clinical professionals who were involved in the procurement of wound care products. Clinical professionals included community nurses, podiatrists, tissue viability or burns specialist nurses, wound research nurses and clinical nurse managers (who had a clinical role, managed a team of community nurses and were responsible for wound product procurement decisions). Non-clinical professionals included: medicines optimisation pharmacists, procurement leads, procurement advisors and medicines	19/441 7/128
------------	---	---	--	---------------------

			management leads. There were five multidisciplinary focus group interviews for clinical professionals; one for each participating provider organisation.	
Reviewer 2	Was the conclusion regarding the importance of academic support based on only one research nurse and one organisation?	This information was not based on one research nurse. As mentioned in the findings, participants from the research active organisation were all actively involved in keeping up to date with research, participating in research and implementing research findings.	Only participants from the provider organisation with a history of collaborative wound care research indicated that they actively sought to keep up to date with research. Specialist nurses from this focus group talked about their established links with university researchers and their involvement in co-producing wounds research with academics. They discussed disseminating relevant research findings through electronic newsletters, workshops and meetings with community staff and where capacity allowed, staff were supported	13/293

			to implement research findings. Participants reported that their organisation was highly research active in wound care...	
Reviewer 2	Please kindly defend the lack of PPI input	PPI input was used to inform some of the questions and prompts for focus group interviews as stated in Lines 185-188. We have removed the first part of this sentence, 'patients and public were not involved in the interviews', to avoid any confusion. As the purpose of the study was to gather views from healthcare professionals and not patients or the public, we realise that we do not need to state that patients were not involved in the interviews.	Views expressed by members of the NIHR CLAHRC Greater Manchester Wounds Research PPI Forum about their experiences with healthcare professionals and wound care services were used to inform some of the questions and prompts for the focus group interviews.	9/185
Reviewer 2	Data analysis methods are unclear especially in regards to comparing and collating data from the two different groups. There is little information on qualitative analysis (I cannot find Figure 1 which may explain this?)	Figure 1 describes the qualitative data analysis process. We compared similarities and differences between participants based on their role, experience and organisational procedures. We apologise that the reviewer was unable to see this Figure. We have checked and it was included as part of our submission.	Data interpretation involved comparing similarities and differences between participants views (depending on factors such as their role, experience and organisational processes).	Figure 1

Reviewer 2	Is there evidence to support the staff views on resources, time available etc or are these staff perceptions?	This study is a qualitative exploration of factors perceived to influence community wound care. Participants reported that their workloads had increased and they	Some participants said they were working more intensely and without breaks, constantly feeling anxious that they may have missed something as time was limited between patient consultations. They reported that there was an increase in sick	11/214
------------	---	---	--	--------

		worked without breaks because there were fewer staff to cover workloads. One reported that there were 30 vacancies in their service. Participants who worked across acute and community services were very aware of the variation in resources	leave, experienced colleagues were leaving and their roles were left vacant. ".....we've got 30 vacancies at the moment that haven't been filled" (Clinical manager) "I just don't feel the acute side has got a grip at all on community services in terms of what we do...I mean, I do a specialist [acute] clinic on a Tuesday morning and have access to all sorts of dressings. And I come back into the community....and we're very limited, we've got one foam [dressing] that we can use." (Podiatrist)	11/220 12/250
--	--	--	---	----------------------

Reviewer 2	As this is based on community wound care – what was the role of the specialist nurses in the focus groups?	The specialist nurses were employed either by a community NHS trust or by a combined community and acute trust. The majority of their work is carried out in the community. They manage their own caseload as well as receiving referrals from community nurses to provide specialist advice and support. They also provide formal training for a range of healthcare professionals. A large proportion of their work involves joint consultations with community nurses and podiatrists. Their role is	In the UK, the management of people with complex wounds is mainly carried out in patients' homes or community clinics by community nurses with advice and support from specialist teams (nurses and medics with expertise in tissue viability, burns, vascular medicine or dermatology).	3/78
------------	--	---	--	------

		outlined in the introduction.		
Reviewer 2	Discussion (lines 483-490) – I am not clear if this is derived from the study findings or is a conclusion?	This is a repetition of the beginning of this paragraph so we have removed it and moved the evidence up to the findings.	Wound care services were described by participants as a working environment characterised by increasing time pressures and diminishing resources. Clinic sessions had been cut, resulting in an increase of home visits for community nurses to non-housebound patients. Roles were perceived as becoming task orientated which was felt to dilute the quality of care. Participants reported there was a rise in sickness, colleagues were leaving for less pressured roles and vacancies were not being filled. UK surveys of community nursing services have found similar results. ³⁻⁵ Championing flexible career pathways, ⁶ integrated care and the introduction of combined hospital and community posts to standardise practice, improve care coordination and vary work experiences have been proposed by UK governing bodies to improve retention rates. ⁷⁻⁹	16/359

Reviewer 2	The discussion could be somewhat shortened in my opinion with respect to the study findings but also needs more links to previous published work.	We have shortened the discussion section. As we mention in the first paragraph, we believe this is to be the first study to explore factors influencing community wound care. Research relevant to the study findings is referenced throughout the discussion.		
------------	---	---	--	--

Reviewer 2	Introduction: comprehensive assessment of the person: please detail what this means	Comprehensive assessment has been defined.	comprehensive assessment of the person and their wound (involving demographics, risk factors for wound healing, quality of life measures, wound status, wound parameters and symptoms)	3/84
Reviewer 2	Introduction: is it worth adding detail of the aims of wound dressings in the introduction – ie what does clinically effective mean in this context and how is this assessed – healing time, pain, infection, exudate??	We believe the introduction adequately sets the context for the study, highlighting both nature and extent of variations in wound care practices. Providing detail on thresholds for effectiveness and cost effectiveness of medical devices and other treatment options increases the word count and is not necessary in this instance.		
Reviewer 2	Introduction: Line 100: do you mean that there is no evidence for dressing type but good evidence for compression therapy? Are there any other treatments with evidence?•	We believe that we have been clear; that there is no evidence that one dressing type is more clinically or cost effective than another, even in the case of relatively expensive anti-microbial dressings. In contrast there are effective first line treatments which should be widely used, such as the use of compression therapy for venous leg ulceration which is known to reduce time to wound healing	Whilst dressings are used widely across wound types, with many different options available for use, there is currently no evidence that one dressing type is more clinically or cost effective than another, even in the case of relatively expensive anti-microbial dressings. In contrast there are effective first line treatments which should be widely used, such as the use of compression therapy for venous leg ulceration which is known to reduce time to wound healing ^{5 6}	4/90
Reviewer 2	Introduction: Line 110 – just compression therapy??	Compression therapy is used as an illustrative example hence the use of ‘such as’	In contrast there are effective first line treatments which should be widely used, such as the use of compression therapy for	4/92

			venous leg ulceration which is known to reduce time to wound healing ^{5 6}	
Reviewer 2	Introduction: Line 121: dressings or all wound care? Is this for complex wounds or all wounds?	This refers to wound care and not just dressings as stated in line 102. The Leading Change, Adding Value Nursing and Midwifery Framework refers to wound care and does not specify whether this is for complex wounds only, which is why we have used the term wound care and not complex wound care.	In the UK, variations in wound care are being recognised and addressed with initiatives such the Leading Change, Adding Value Nursing and Midwifery Framework	4/102
Reviewer 2	Introduction: Is there an aim to assess the link to University support? This is mentioned in the methods?	No, our aim was to identify factors that influence care in community settings for patients with complex wounds.	Our aim was to identify and explore factors that influence care in community settings for patients with complex wounds. We wanted to better understand the current context of community wound care and how research evidence informs care delivery.	4/108
Reviewer 2	Introduction: I think the last sentence should be in Methods and leave the aim as the last intro sentence	We have removed this sentence from the introduction but not added it to the methods as we feel that the eligibility criteria is adequately explained.		
Reviewer 2	Methods: Line 130: as above – complex or all wounds?	We have amended the sentence to clarify	We conducted six focus group interviews to explore the factors that influence the care of people with complex wounds in community settings.	5/117
Reviewer 2	Methods: Line 159: why did they differ and in what way? Which format is referred to in line 159?	They differed because their roles were different; clinical professionals provided clinical care for patients with complex wounds whereas procurement staff managed procurement and procurement systems.	As the themes for clinical and non-clinical focus group interviews differed, we chose to separate clinical from non-clinical professionals to maintain focus and create an optimum environment for free flowing discussions.	7/140 Appendix 1 Appendix 2

		The interview schedules therefore differed as presented in Appendix 1 and 2. We have changed the wording to explain the format used.		
Reviewer 2	Methods: Line 167-171 seems to contradict itself?	We have amended the wording to clarify	As participants were drawn from a relatively homogeneous population and the interview schedules were focused on specific aspects of wound care and wound product procurement, we anticipated that we would reach data saturation within three to four focus group interviews, however, to incorporate all partner provider organisations using the format described above we needed to recruit 50-60 participants in total across the six groups (to allow for 8 to 10 participants per group), based on recommendations from existing literature.	7/147

Reviewer 2	Methods: Please add details on assessing the link to University support, defining complex wounds for participants, defining wound care	As mentioned above, the aim was to identify factors that influence care in community settings for patients with complex wounds. We have defined complex wounds in the introduction and have amended the wording to define complex wound care.	People with complex wounds (open wounds, such as foot, leg and pressure ulcers, burns, open trauma and surgical wounds that are difficult to heal), Care of complex wounds in community settings normally involves a comprehensive assessment of the person and their wound (involving demographics, risk factors for wound healing, quality of life measures, wound status, parameters and symptoms), specific wound-related assessments such as ankle brachial pressure index (ABPI) for people with venous leg ulcers and implementation of appropriate interventions. ⁴ Interventions may involve wound cleansing followed by dressing to manage exudate and protect the wound.	3/77 3/84
------------	--	--	---	------------------

Reviewer 2	Methods: Data collection: how did the questions vary between the two groups and how were the data from the two groups collated? How was the data formally linked to the TDF domains?	Appendix 1 and 2 present the interview schedules for clinical and non-clinical focus group interviews. Deductive coding by two researchers independently, was used to match themes to the 14 TDF domains.		Appendix 1 Appendix 2
Reviewer 2	Methods: What were the themes for the interviews – was it completely open at the start – eg was photography always asked about, time for dressing / wound care?? Was the topic guide modified for subsequent interviews?	The clinical focus group interview schedules remained the same for each interview. The non-clinical focus group interview schedule was different and focused on procurement of wound care products and procurement processes.		Appendix 1 Appendix 2
Reviewer 2	Methods: Please justify why patients were not involved in any way – could they not have been involved to assess questions asked?	The CLAHRC PPI wounds forum was in the process of being formed at the time of this study, early discussions about their wound care experiences were used to inform some of the questions.	Views expressed by members of the NIHR CLAHRC Greater Manchester Wounds Research PPI Forum about their experiences with healthcare professionals and wound care services were used	9/185

			to inform some of the questions and prompts for the focus group interviews.	
--	--	--	---	--

Reviewer 2	Methods: Data analysis – how were the themes extracted and how were the two groups compared? Was a software package used?	Transcripts and additional field notes were stored, coded and analysed in Nvivo 11. Inductive coding established the main themes and then deductive coding by two researchers independently, was used to match themes to the 14 TDF domains. An analytical framework was developed and organised into themes, sets and cases for comparison. This information is presented in Figure 1.	Comprehensive Familiarisation Process was conducted by one researcher (TG) to contextualise the material. This involved proofreading and cross-checking against the recordings for accuracy and de-identification. Transcriptions were annotated with additional field notes and stored in NVivo 11. Inductive Coding was carried out by TG to establish the main themes. Deductive coding was undertaken by both researchers independently, to apportion themes to domains, adding subdomains as required to ensure that important data were not omitted. Regular meetings facilitated critical exploration of participant responses and agreement on domain definitions in the context of the data set. An Analytical framework was developed by two researchers (TG and PW). The thematic codes generated inductively were aligned with the 14 theoretical domains of the TDF. A framework matrix was developed to reduce and organise data into themes, cases and sets for ease of comparison. Data interpretation involved comparing similarities and differences between participants' views (depending on factors such as their role, experience and organisational processes) and mapping connections between categories to explore relationships and/or causality.	Figure 1
Reviewer 2	Methods: I cannot find Figure – apologies iof I have missed it?	Figure 1 was submitted with the previous revision and has been resubmitted for this revision.		

Reviewer 2	Results: Why only one research nurse?	In general it is important to note the community-focused research nurses are less common than in acute settings. The research active organisation involved was the only organisation to employ community wound research nurses who worked closely with community healthcare professionals such as district nurses, specialist nurses and podiatrists which limited the number of research nurses who could attend. The attending research nurse was from a small team of clinical researchers. We felt having one representative from this group was adequate representation from this small group.	Only one research nurse was able to participate and as the research-active organisation was the only organisation to employ a small team of wound research nurses, it is not surprising that we could only recruit one.	20/452
Reviewer 2	Is there evidence for understaffing (line 274) or is this a perception?	Participants reported that clinics that were managed by six or seven specialist staff were now run by two (non-specialist) community nurses	“Physically running the clinic was based on when there was about six or seven [leg ulcer specialist] staff ... when it was a leg ulcer service. There's only two of us so we haven't got the capacity to cover those let alone do all the home visits.” (Community nurse)	11/225
Reviewer 2	A similar question regarding line 281?	Participants stated that they wasted time gathering information that hospital staff should have provided with the referral information.		
Reviewer 2	Lines 411-413: I am not clear what this means?	We have amended the paragraph to clarify.	As referred to above, all participants were concerned about the influence that the pharmaceutical industry had on product choices. It was felt that this influence varied depending on how closely pharmaceutical representatives' access was monitored	14/319

Reviewer 2	Line 437: what does 'mis-use' mean?	This refers to overuse of antimicrobials without close monitoring. We have changed misuse to overuse	Community nurses reported that antimicrobial dressings (particularly silver-impregnated dressings) were used for individual patients for a two-week trial period and then reviewed, however, they acknowledged that if use was not closely monitored there was potential for overuse .	15/336
Reviewer 2	Discussion: Line 465: for complex wounds alone?	We have amended the sentence to clarify.	We believe this is the first study to explore factors influencing care in community settings for people with complex wounds whilst seeking to understand the reasons for known variation in practice.	16/351

Reviewer 2	Discussion: Lines 483-490: is this a conclusion or does this come from the results?	We have amended the sentence to clarify	Wound care services were described by participants as a working environment characterised by increasing time pressures and diminishing resources. Roles were perceived as becoming task orientated which was felt to dilute the quality of care. Participants reported there was a rise in sickness, colleagues were leaving for less pressured roles and vacancies were not being filled. UK surveys of community nursing services have found similar results. ³⁵ The UK has fewer nurses relative to the population than many EU countries. ¹⁰ The number of community nurses is falling, with an estimated vacancy rate of 9.4%. ¹¹ Forty percent of experienced nurse positions are vacant. ⁸ Championing flexible career pathways, integrated care and the introduction of combined hospital and community posts (to standardise practice, improve care coordination and vary work experiences) have been proposed by UK governing bodies to improve retention rates. ^{6 7 9 12}	16/359
------------	---	---	---	--------

Reviewer 2	Discussion: Was the 'restrictive formulary' based on evidence?	Yes, formularies guide practice by restricting prescribers' choices. NICE (2014) recommends the development of formularies to improve patient outcomes by optimising prescribing choices based on patient factors, to reduce inappropriate variation in care and support prescribers to follow guidance by professional regulatory bodies. ¹³ Participants discussed the formularies that were in place in their organisation and those who used more restrictive formularies found the limited choice enabled more guided decision making as stated in the findings (lines 263270). The NICE reference has been added.	Participants reported a variety of wound care product procurement processes; some (across two provider organisations) obtained all products via prescription, others (across two provider organisations) used a combination of prescribing and stock purchase and one group (one organisation) operated a total stock purchase system. All participants noted the local use of wound care formularies (a locally developed list of recommended products), to guide prescribing or purchasing decisions, ¹³ however, through discussion it was recognised that the products listed and the number of product available varied across formularies. One organisation had a very restrictive formulary and monitored use closely; participants found this restrictive formulary enabled them to choose appropriate products.	12/263
Reviewer 2	Discussion: Lines 579-581: it may not be surprising but what bias would this confer to the findings	We have amended the text to add detail as described above.	Only one research nurse was able to participate and as the research-active organisation was the only organisation to employ a small team of wound research nurses it is not surprising that we could only recruit one. ^{4 14-17}	20/452
Reviewer 2	Conclusion: what does: 'are perceived' mean? By whom?	Wording amended to clarify.	Our study provides new insight into the role experiential learning and social influences play in determining management and treatment choices and on the limited influence of evidence obtained from research. Workforce pressures and limited resources are perceived by the participants to impede care by reducing patient access to services, the ability to provide holistic care.	20/473

Reviewer 2	Conclusion: The link between university and health care organisations is not in the methods and therefore shouldn't be in the conclusion? Is this based on one participant?	As mentioned above, the link between university and healthcare organisations is a potential factor along with other potential factors that we wanted to understand better. Also as mentioned above, the conclusion is not based on one participant. The majority of the discussions around the use of research was driven by the specialist nurses, podiatrists and community nurses and not the research nurse.	Only participants from the provider organisation with a history of collaborative wound care research indicated that they actively sought to keep up to date with research. Specialist nurses from this focus group talked about their established links with university researchers and their involvement in co-producing wounds research with academics. They discussed disseminating relevant research findings through electronic newsletters, workshops and meetings with community staff and where capacity allowed, staff were supported to implement research findings. Participants reported that their organisation was highly research active in wound care;	13/293
Reviewer 3: Andrea Patey	I believe the authors have adequately responded to concerns based by myself and the other reviewers.	Thank you for your comments		

References

1. Coleman S, Nelson EA, Vowden P, et al. Development of a generic wound care assessment minimum data set. *J Tissue Viability* 2017;26(4):226-40. doi: 10.1016/j.jtv.2017.09.007
2. Ashby RL, Gabe R, Ali S, et al. Clinical and cost-effectiveness of compression hosiery versus compression bandages in treatment of venous leg ulcers (Venous leg Ulcer Study IV, VenUS IV): a randomised controlled trial. *Lancet* 2014;383(9920):871-9. doi: 10.1016/S0140-6736(13)62368-5
3. Ball J, Philippou J, Pike G, et al. Survey of district and community nurses in 2013: report to the Royal College of Nursing. London: Royal College of Nursing, 2014.
4. Maybin J, Charles A, Honeyman m. Understanding quality in district nursing services: Learning from patients, carers and staff. London: The King's Fund, 2016.
5. National Quality Board. Safe, sustainable and productive staffing; An improvement resource for the district nursing service 1ed. London: National Quality Board 2017.
6. Public Health England. Facing the Facts, Shaping the Future: A draft health and care workforce strategy for England to 2027. London: Public Health England, 2017.
7. Goodwin N, Sonala L, Thiel V, et al. Co-ordinated care for people with complex chronic conditions: Key lessons and markers for success. London: The King's Fund, 2013.
8. NHS England. Framework for commissioning community nursing. London: NHS England, 2015.
9. NICE. Transition between inpatient hospital settings and community or care home settings for adults with social care needs Nice guideline [NG27] National Institute for Health and Clinical Excellence 2015.
10. Buchan J, Charlesworth A, Gershlick B, et al. Rising pressure: the NHS workforce challenge: Workforce profile and trends of the NHS in England. London: The Health Foundation 2017.

11. OECD. The Nursing Workforce Health Statistics: Organisation for Economic Co-operation and Development 2017.
12. HEE. Raising the Bar. Shape of Caring: A Review of the Future Education and Training of Registered Nurses and Care Assistants. London: Health Education England, 2018.
13. NICE. Developing and updating local formularies [MPG1]. Medicines practice guideline: National Institute for Health and Care Excellence, 2014.
14. Robertson R, Wenzel L, Thompson J, et al. Understanding NHS financial pressures How are they affecting patient care? London: The King's Fund, 2017.
15. Jackson C. Implementing a District and Community Nursing workload tool, to determine safe staffing levels and skill mix in a community care provider organisation: An economic assessment of potential benefits for workforce planning. Kent: Canterbury Christ Church University 2016.
16. Saltman RB. The impact of slow economic growth on health sector reform: a cross-national perspective. *Health Econ Policy Law* 2018;1-24. doi: 10.1017/S1744133117000445
17. Karanikolos M, Mladovsky P, Cylus J, et al. Financial crisis, austerity, and health in Europe. *Lancet* 2013;381(9874):1323-31. doi: 10.1016/S0140-6736(13)60102-6